# Dietary cholesterol promotes steatohepatitis related hepatocellular carcinoma through dysregulated metabolism and calcium signaling

Jessie Qiaoyi Liang[1], Narcissus Teoh[2], Lixia Xu[1], Sharon Pok[2], Xiangchun Li[1], Eagle S.H. Chu[1], Jonathan Chiu[1], Ling Dong[3], Evi Arfianti [2], W. Geoffrey Haigh[4], Matthew M. Yeh[5], George N. Ioannou[4], Joseph J.Y. Sung[1], Geoffrey Farrell[2] & Jun Yu [1]

The underlining mechanisms of dietary cholesterol and nonalcoholic steatohepatitis (NASH) in contributing to hepatocellular carcinoma (HCC) remain undefined. Here we demonstrated that high-fat-non-cholesterol-fed mice developed simple steatosis, whilst high-fat-high-cholesterol-fed mice developed NASH. Moreover, dietary cholesterol induced larger and more numerous NASH-HCCs than non-cholesterol-induced steatosis-HCCs in diethylnitrosamine-treated mice. NASH-HCCs displayed significantly more aberrant gene expression-enriched signaling pathways and more non-synonymous somatic mutations than steatosis-HCCs (335 ± 84/sample *vs* 43 ± 13/sample). Integrated genetic and expressional alterations in NASH-HCCs affected distinct genes pertinent to five pathways: calcium, insulin, cell adhesion, axon guidance and metabolism. Some of the novel aberrant gene expression, mutations and core oncogenic pathways identified in cholesterol-associated NASH-HCCs in mice were confirmed in human NASH-HCCs, which included metabolism-related genes (*ALDH18A1, CAD, CHKA, POLD4, PSPH and SQLE*) and recurrently mutated genes (*RYR1, MTOR, SDK1, CACNA1H and RYR2*). These findings add insights into the link of cholesterol to NASH and NASH-HCC and provide potential therapeutic targets.

[1] Department of Medicine and Therapeutics, State Key Laboratory of Digestive Disease, Institute of Digestive Disease and Li Ka Shing Institute of Health Sciences, CUHK Shenzhen Research Institute, The Chinese University of Hong Kong, Hong Kong, China. [2] Liver Research Group, Australian National University Medical School at the Canberra Hospital, Garran, ACT, Australia. [3] Department of Gastroenterology and Hepatology, Zhongshan Hospital of Fudan University, Shanghai, China. [4] Department of Gastroenterology and Hepatology, Veterans Affairs Puget Sound Health Care System and University of Washington, Seattle, WA, USA. [5] Department of Pathology, University of Washington School of Medicine, Seattle, WA, USA. These Co-first author: Narcissus Teoh.   Correspondence and requests for materials should be addressed to G.F. (email: geoff.farrell@anu.edu.au) or to J.Y.  (email: junyu@cuhk.edu.hk)

Hepatocellular carcinoma (HCC) is the second leading cause of cancer death in men and sixth in women worldwide[1]. China alone accounts for about 50% of the total case burden and deaths worldwide[1]. While hepatitis B and C viruses (HBV and HCV) remain the most important risk factors, virus-related HCC is expected to decrease in the near future due to widespread adoption of HBV vaccination and curative HCV treatments. However, the prevalence of obesity and type 2 diabetes is increasing. This is associated with an increase in non-alcoholic fatty liver disease (NAFLD), including its histologically progressive form, non-alcoholic steatohepatitis (NASH)[2]. Strong evidence has emerged that obesity and NASH are major risk factors for HCC[3,4].

NAFLD is the most common liver disease, affecting 20–40% of adults[5,6]. The spectrum of liver pathology extends from simple steatosis, in which the only feature is excessive fat deposition within hepatocytes, to NASH, in which additional features include hepatocyte injury, liver inflammation and pericellular fibrosis. Ten-30% of patients with NAFLD have NASH[5]; the accompanying fibrosis may progress to cirrhosis and HCC[7]. Further, there is growing evidence from case series that HCC can occur in patients with non-cirrhotic NASH.

Dietary cholesterol has been shown to play a role in the development of steatohepatitis in both animal models and human, and in epidemiological studies cholesterol intake is an independent risk factor for HCC[8–10]. However, the mechanism by which cholesterol promotes NASH and leads to HCC development is unclear, and it has also not been established whether the resultant HCCs harbour a different set of genetic mutations and other molecular changes that differs from HCCs in other etiopathogenic settings.

In the present study, we explored the role of dietary cholesterol in contributing to NASH and NASH-HCC development in mice fed high-fat diets with or without high cholesterol. We found that animals fed high-fat with high cholesterol (HFHC) showed NASH development, whereas animals fed high-fat without cholesterol (HF diet) developed simple steatosis. HCCs were more numerous and larger in HFHC-fed than in HF-fed mice exposed to diethylnitrosamine (DEN). Paired tumor and adjacent non-tumorous liver from mice fed HFHC vs. HF diets were analyzed by expressional profiling and whole-exome sequencing to seek aberrant gene expression, genomic mutations and signaling networks that could be enriched in the development of dietary cholesterol-induced liver tumors. Finally, to establish the clinical relevance of the discoveries in murine models, we searched for similar, disease-specific gene expression changes and somatic mutations in human NASH-HCCs.

## Results

**Dietary cholesterol causes NASH in mice fed high-fat diet**. To explore the effects of dietary cholesterol on fatty liver disease, we fed mice either normal chow, HF, or HFHC diets (Fig. 1a). HF- and HFHC-fed mice gained more weight than controls fed normal chow (both $P < 0.001$) (Fig. 1b). Likewise, both HF- and HFHC-fed mice were hyperglycemic (both $P < 0.001$) and displayed impaired glucose tolerance (both $P < 0.05$) compared to control mice (Fig. 1c). Although HF- and HFHC-fed mice were similarly obese, HFHC-fed mice showed significantly higher liver weight and liver/body weight ratios (both $P < 0.001$) (Fig. 1d). Further, HFHC-fed mice showed significantly increased hepatic triglyceride, free cholesterol and cholesterol ester contents, as well as serum cholesterol compared to HF-fed mice (Fig. 1e). Histologically, only steatosis was found in the non-tumorous liver of HF-fed mice, while those from HFHC-fed mice showed steatosis, hepatocyte ballooning and inflammatory cell infiltration sufficient

to be deemed NASH (Fig. 1f). Accordingly, steatosis and lobular inflammation scores were significantly higher in HFHC vs. HF livers, contributing to significantly higher NAFLD activity scores in HFHC livers (all $P < 0.05$; Fig. 1g).

**Dietary cholesterol augments hepatocarcinogenesis**. To explore the role of dietary cholesterol in HCC development, we compared DEN-induced hepatocarcinogenesis in mice fed normal chow, HF, or HFHC diets. HCC incidence was numerically higher in HF-fed mice (90%) and HFHC-fed mice (100%) compared with normal chow-fed controls (67%). Impressively, HCC multiplicity and size were markedly increased in HFHC-fed versus HF-fed mice ($P < 0.0001$; Fig. 1h). Furthermore, 40% of HCCs in HFHC-fed mice but none in HF-fed mice showed lung metastases (Supplementary Fig. 1). These findings demonstrate that high dietary cholesterol augments high-fat diets in promoting hepatocarcinogenesis, in association with inducing NASH live pathology.

**Dietary cholesterol upregulated inflammatory genes**. In light of the effect of cholesterol on NASH development, we compared the gene expressional profiles of NASH livers from HFHC-fed mice and steatosis livers from HF-fed mice. We identified 634 upregulated genes and 248 downregulated genes (≥2 or ≤−2 fold, $P < 0.05$) (Fig. 2a). Kyoto Encyclopedia of Genes and Genomes (KEGG) pathway enrichment analysis revealed that, in HFHC-fed mouse liver, aberrant gene expression was mainly enriched in inflammation, metabolism and cancer-related pathways (Fig. 2b). The affected inflammatory pathways ranged from macrophage infiltration (upregulation of macrophage markers), early inflammatory response (signaling of cytokines, chemokines and interleukins), late-phase fibrotic response to NASH, and transforming growth factor (TGF)-β and Wnt-related signaling (Fig. 2c, d). Dysregulation of metabolism and cancer-related pathways (metabolic pathways, pathways in cancer, calcium signaling, insulin signaling, cell adhesion, axon guidance, etc) could possibly function in promoting malignant transformation of HFHC-livers (Supplementary Fig. 2). These findings demonstrate the necro-inflammatory nature of gene expression changes in liver in response to HFHC diet that might be associated with NASH and with accelerated development of HCC.

**Aberrant gene expression in murine NASH- and steatosis-HCCs**. To understand the molecular basis for accentuated hepatocarcinogenesis in HFHC-fed vs. HF-fed mice, we compared gene expression profiles of NASH-HCC with steatosis-HCC directly. We noted that 4,660 genes were aberrantly expressed (2-fold or more) in NASH-HCC versus steatosis-HCC. Because many of these genes could include those pertinent to the disease process of NASH versus steatosis, we further analyzed the differential gene expression in HCCs as compared to adjacent non-tumorous livers, and then compared cancer-related expressional changes between the two groups of HCCs. Similar numbers of aberrantly expressed genes were identified in NASH-HCCs and steatosis-HCCs, and more genes were upregulated (315 and 364 respectively) than downregulated (40 and 47, respectively) in both groups (Fig. 3a). Cancers that developed between the two dietary groups shared about half (164) of upregulated genes. These were enriched in 15 pathways (Fig. 3b), including pathways important in cancer (ErbB signaling, MAPK signaling, PPAR signaling, etc), necro-inflammation (TGF-β signaling, chemokine signaling, etc) and cellular metabolism. More pathways were specifically dysregulated in NASH-HCCs, including those previously reported in NAFLD (calcium, insulin, hedgehog and adipocytokine signaling[11,12]), and cancer (axon guidance and cell adhesion). Notably, most of the pathways dysregulated in NASH-HCCs (vs.

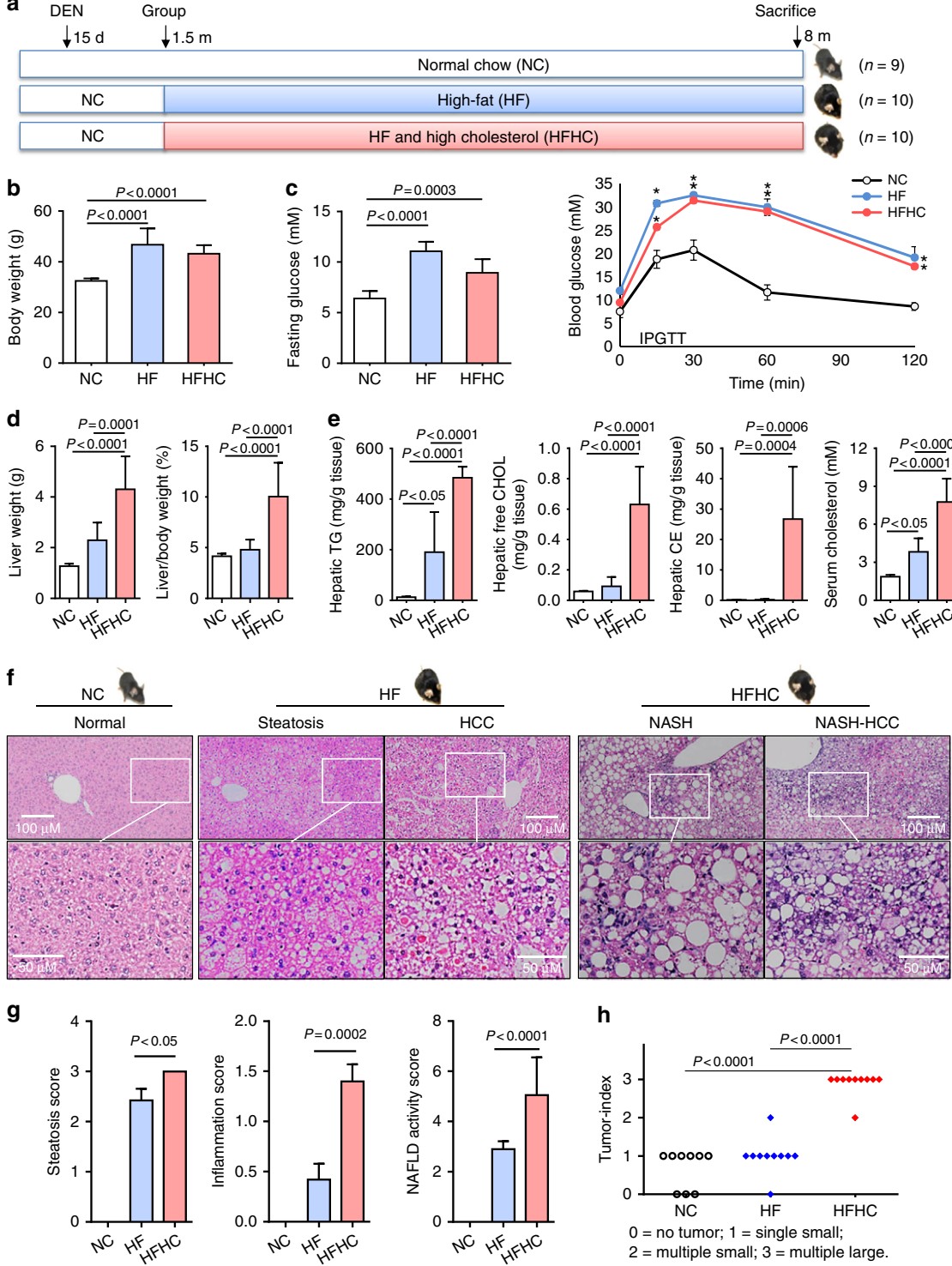

**Fig. 1** Cholesterol augmented high-fat (HF) diet in accelerating hepatocarcinogenesis in mice. **a** Schematic illustration of the treatment of mice. Male C57BL/6J mice were administered a single injection of the carcinogen diethylnitrosamine (DEN) intraperitoneally at age 15 days and fed normal chow ($n = 9$), HF ($n = 10$) or high-fat high-cholesterol (HFHC; $n = 10$) diets starting from 6 weeks of age till 8 months. **b** Body weights of mice fed HF or HFHC diets were similar and both significantly higher than normal chow (NC)-fed mice. **c** Fasting blood glucose and intraperitoneal glucose tolerance test (IPGTT) levels in mice fed NC, HF or HFHC diets. $*P < 0.05$. **d** Mice fed HFHC diet had significantly higher liver weights and liver-body weight ratios than HF-fed mice. **e** Levels of hepatic triglyceride, hepatic free cholesterol and cholesterol ester, and serum cholesterol were measured in HF- and HFHC-fed mice. **f, g** Representative H&E staining histological images (**f**) and NAFLD scores (**g**) of liver tissues from HF- and HFHC-fed mice. **h** HCC incidence, number and size in NC-, HF- and HFHC-fed mice after DEN injection. Tumor number: few < 5; multiple ≥ 5. Tumor size: small, all < 5 mm³; large, at least one ≥ 5 mm³. The data are shown as means ± SE for IPGTT and means ± SD for others. Data in b-g between each two groups were compared using ANOVA Tukey's multiple comparison tests

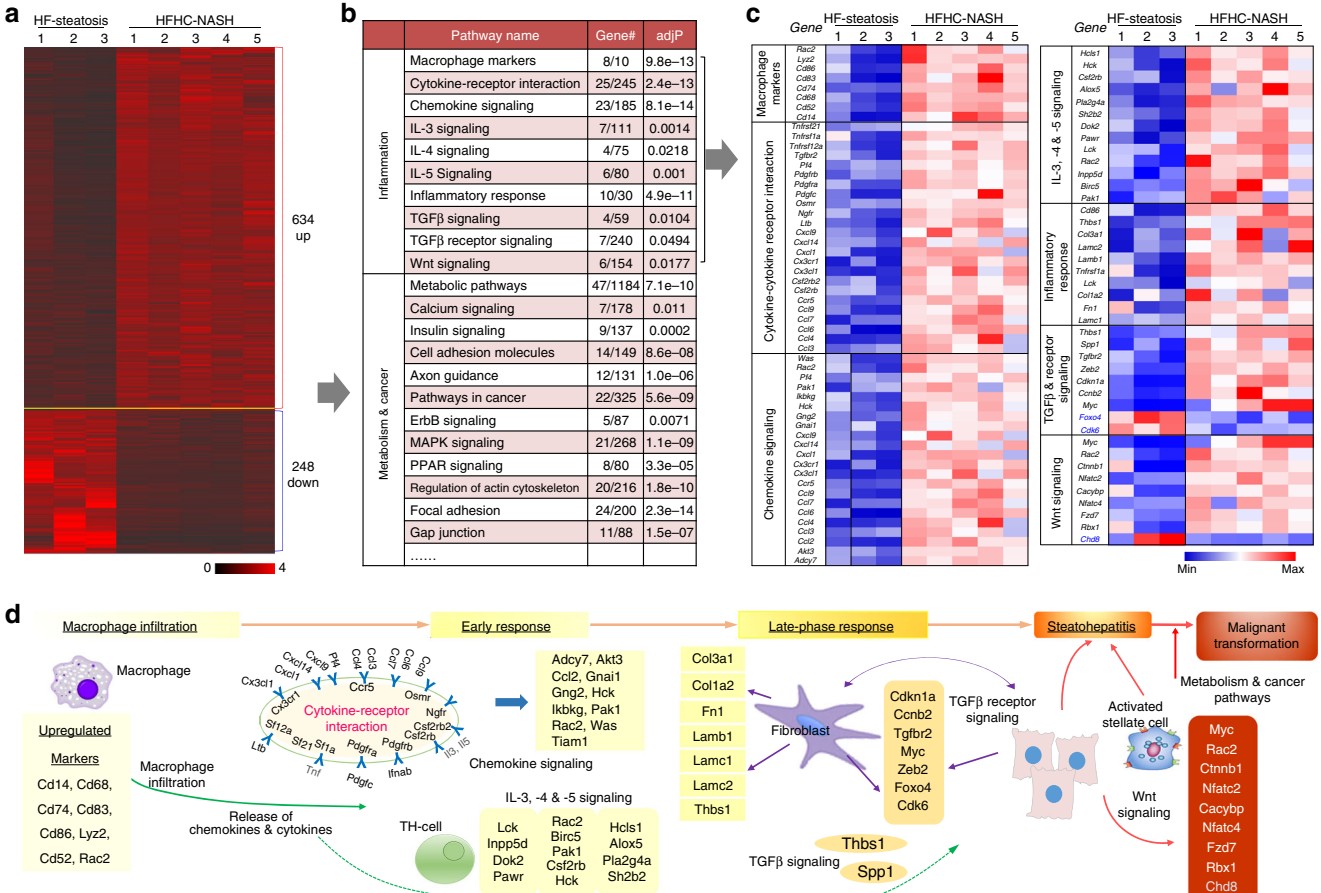

**Fig. 2** Expressional aberration of inflammation-related genes associated with NASH development in HFHC-fed mice. **a** Clustering of differentially expressed genes in HFHC-induced NASH as compared to HF-induced steatosis. **b** Pathways enriched by differentially expressed genes in NASH vs. steatosis. #gene, the number of annotated genes in the input list / number of annotated genes in the reference list. adjP, $p$ value adjusted by multiple test adjustment. **c** Differentially expressed inflammatory genes in NASH compared to steatosis. Expression levels were normalized to the mean level of each gene among all samples shown. **d** Schematic illustration of the dysregulated pathways involved in the development of HFHC-induced NASH

surrounding livers) were already dysregulated in NASH (vs. steatosis) by aberrant gene expression (Figs. 2b, 3b). These include calcium signaling, pathways in cancer, ErbB signaling, MAPK signaling, and chemokine signaling, metabolic pathways etc. Genes commonly upregulated in both NASH- and steatosis-HCCs when compared to adjacent non-tumorous liver (such as *Cdh1, Pak1, Sqle*, and others) may be important in hepatocarcinogenesis. We also identified genes aberrantly expressed only in NASH-HCCs, such as downregulation of *Cfd* (complement factor D, encoding adipsin), and upregulation of *Ddit3, Itga6* and others (Fig. 3c). These findings implicate sets of genes and pathways specifically associated with the acceleration of hepatocarcinogenesis occasioned by addition of cholesterol to a HF diet through expressional dysregulation.

**Aberrant gene expression verified in human NASH-HCCs.** To establish whether the present dietary model of NASH-enhanced hepatocarcinogenesis has relevance to human disease, we determined whether genes aberrantly expressed in mouse NASH-HCCs (Fig. 3c) were also differentially expressed in human HCCs obtained from patients with histologically pedigreed NASH. Among the 17 available human NASH-HCCs, we verified 12 genes to be significantly up- or downregulated in NASH-HCCs compared to their surrounding liver by RNA sequencing (all $P < 0.05$ by paired $t$ test and false discovery rate (FDR) < 0.15; Fig. 4a). The adipsin-encoding gene *CFD* was significantly downregulated in NASH-

HCCs. Upregulated genes included two cell adhesion molecules (*ALCAM* and *ITGA6*), 3 MAPK signaling genes (*DDIT3, MAP3K6* and *PAK1*) and 6 other metabolic genes (*ALDH18A1, CAD, CHKA, POLD4, PSPH, SQLE*). Differential expression of seven of these genes (*CFD, ALDH18A1, CHKA, DDIT3, ITGA6, PSPH* and *SQLE*) was replicated in another set of 12 paired NASH-HCCs and adjacent non-tumor livers by RT-qPCR (all $P < 0.05$ by paired $t$ test; Fig. 4b). The generalizability of these particular expressional changes between species provides evidence of their likely etiopathogenic importance in HCC related to NASH.

**NASH-HCCs harbor more mutations than steatosis-HCCs in mice.** To identify genomic alterations associated with HCC development in HFHC- and HF-fed mice, whole-exome sequencing was performed. The somatic mutation spectra of both experimental groups were similar, with no observed differences in nucleotide base changes. However, NASH-HCCs harbored significantly more somatic mutations than steatosis-HCCs (circos illustration in Fig. 5a). The average mutated gene numbers were $452 \pm 119$ vs. $58 \pm 20$ in total somatic mutations, and $335 \pm 84$ vs. $43 \pm 13$ in non-synonymous somatic mutations (both $P < 0.05$; Fig. 5b; Supplementary Data 1). Only 28 mutated genes overlapped between NASH- and steatosis-associated HCCs. These findings demonstrate distinct genetic alterations in HCCs related to NASH produced by HFHC diet vs. HCCs in mice consuming only a HF diet. Differences between the two groups

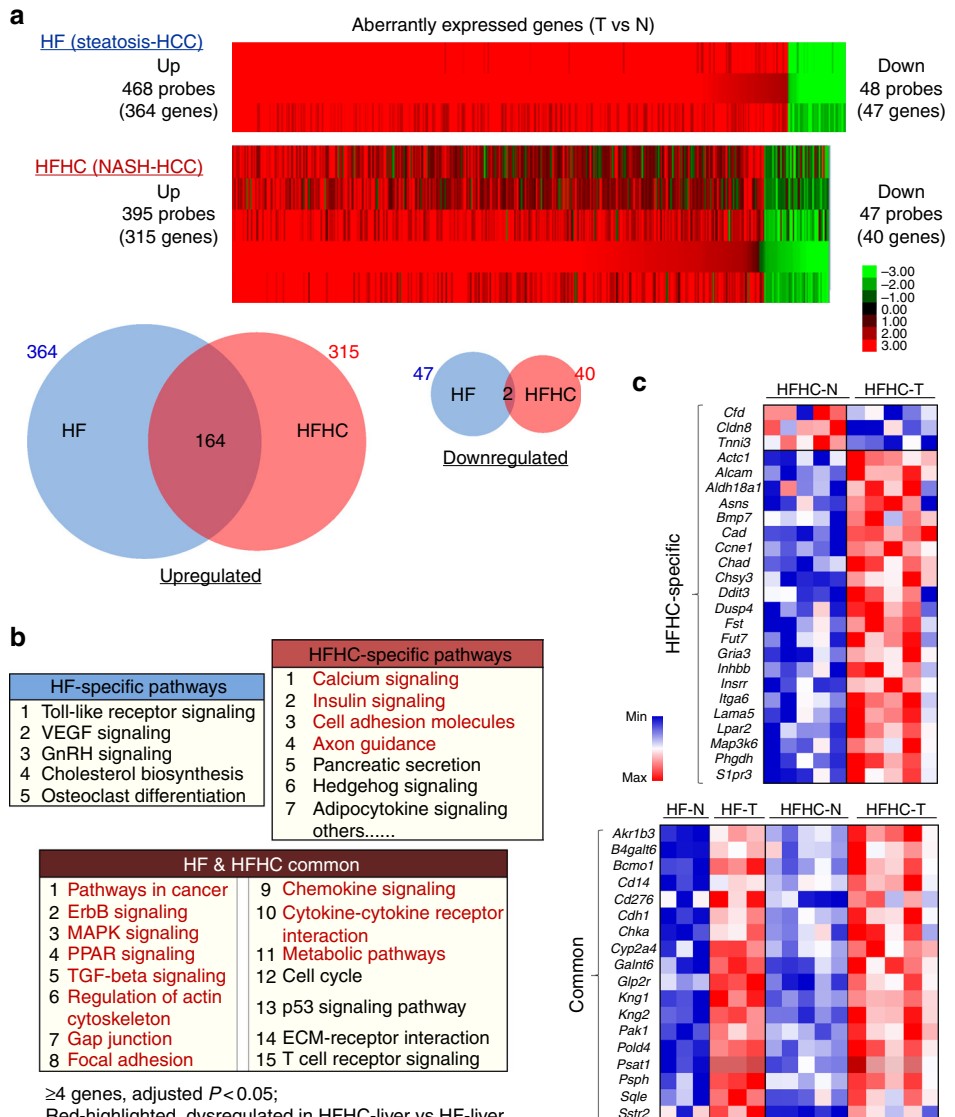

**Fig. 3** Aberrant gene expression and the dysregulated pathways in HCCs from HF- and HFHC-fed mice. **a** Heat map of differentially expressed genes in HCCs versus adjacent non-tumorous samples in HF- and HFHC-fed mice. With the majority in both groups upregulated, the two groups shared about half of the upregulated genes. **b** Pathways dysregulated commonly in both HFHC-associated NASH-HCCs and HF-associated steatosis-HCCs, or specifically in one group, by aberrant gene expression (≥4 genes involved, $P < 0.05$ by multiple test adjustment) are shown. Pathways highlighted in red were also dysregulated in NASH versus steatosis. **c** Differentially expressed genes in HCCs compared to adjacent non-tumorous livers (upper panel, unique in HFHC-fed mice; lower panel, common in both groups). Expression levels were normalized to the mean level of each gene among all samples shown

were evident both in the number of genes mutated and the particular genes affected.

**Identification of mutations related to NASH-HCC in mice.** Several of the mutated genes we identified in murine NASH-HCCs have been reported as drivers in other human cancers, such as *Mtor*[13], *Sdk1*[14], *Braf*[15], *Pik3cb*[16], *Ctnnd1*[17], *Ctnna3*[18], *Akt3*[19], and *Nos2*[20]. We identified 82 genes to be recurrently mutated in two or more NASH-HCCs (Fig. 5c; Supplementary Data 2); these included two genes mutated in 4/5 NASH-HCCs (*Ryr1* and *Sdk1*); 8 genes mutated in 3/5 NASH-HCCs (*Epha8*, *Pcdh15*, *Fat2*, *Cep152*, *Ttn*, *Rxfp1*, *Aox3l1* and *Pkd1l3*) and 71 genes mutated in 2/5 NASH-HCCs (notably *Mtor*, *Ryr2*, *Cacna1h*, *Col7a1*, *Fcgbp*, *Adam29*, *Gpr98* and *Pclo*). Only 2 of the 82 genes (*Ttn* and *Obscn*) were also mutated in one HF-associated steatosis-HCC each, and only one gene (*Kif3c*) was recurrently mutated in steatosis-HCCs. These results suggested that mutations of these known or potential cancer

driver genes contribute to the development of cholesterol-associated NASH-HCC. The recurrent mutations suggest that *Mtor* and *Sdk1* may also function as cancer drivers in NASH-HCC, while other recurrently mutated genes could represent novel candidates associated with NASH-HCC.

**Pathways enriched by gene mutations in NASH-HCC in mice.** To investigate the signaling networks enriched by mutations in NASH- and steatosis-HCCs, we conducted KEGG pathway analysis. Seven pathways were found to be significantly enriched by gene mutations in three or more NASH-HCCs, whereas no pathway was significantly affected in steatosis-HCCs (Fig. 6a and Supplementary Data 3). Pathways affected in NASH-HCCs include metabolic, insulin and calcium signaling, cell adhesion and tight junction molecules, ABC transporters, and axon guidance. It is of particular interest that gene mutations expected to dysregulate calcium signaling were present in all five dietary cholesterol-associated NASH-

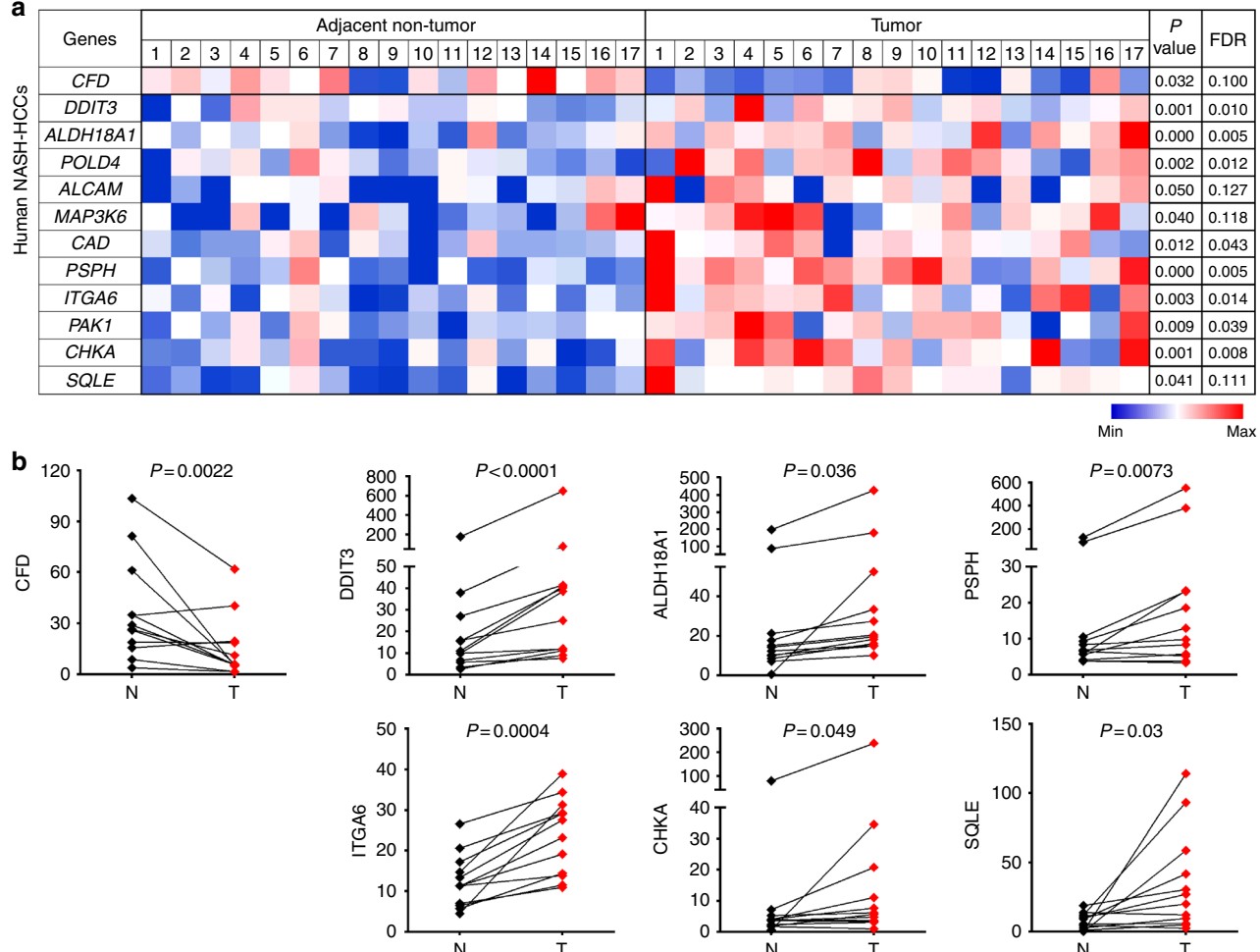

**Fig. 4** Aberrant gene expression verified in human NASH-HCC samples as compared to adjacent non-tumor livers. **a** Shown are RNA levels in FPKM in 17 pairs of tumors compared to adjacent non-tumor livers from patients with NASH-HCC. *P* values by paired *t* tests and FDR adjusted by Benjamini–Hochberg method are shown. Expression levels were normalized to the mean level of each gene among all samples shown. **b** Expression of seven genes was further validated to be significantly altered in 12 pairs of NASH-HCCs (T) compared to adjacent non-tumor livers (N) by RT-qPCR. Comparison by paired *t* tests

HCCs (Supplementary Table 1) as calcium signaling has been reported to be directly affected by cellular cholesterol content[21,22] and plays an important role in cancer development[23,24]. Twenty-eight calcium signaling genes were mutated in NASH-HCCs, with 6 recurrently mutated (*Ryr1, Ryr2, Cacna1d, Cacna1h, P2rx1* and *Itpr1*) all encoding calcium channel proteins, and *Ryr1* was mutated in 4/5 samples (Fig. 6b). In NASH-HCCs, the 16 mutated genes involved in insulin signaling included insulin receptor (*Insr*) and some other well-known cancer-related genes (*Mtor, Hras1, Akt3*, etc), with *Mtor* and *Hras1* recurrently mutated (Fig. 6c). The 75 mutated genes involving in metabolism would be expected to dysregulate 15 specific metabolic signaling pathways, including 6 associated with lipid metabolism (Supplementary Fig. 3b and Supplementary Table 2).

**Dysregulated pathways by mutations and aberrant expression.** Only 4.4% of mutated genes were differentially expressed in NASH-HCCs as compared to adjacent non-tumorous liver (Supplementary Fig. 4). This result indicates that genetic alteration contributed little effect to expressional aberration in NASH-HCC. However, five pathways were commonly and uniquely enriched by mutations and aberrant gene expression in NASH-HCCs; these include calcium, insulin signaling, cell adhesion, axon guidance and metabolic pathways (Fig. 6d). Furthermore, these five pathways were

significantly enriched by aberrant gene expression in HFHC-induced NASH as compared to HF-induced steatosis (Fig. 2b). These findings indicate that dysregulation of the five core pathways is involved in the transition of steatosis to NASH (by aberrant gene expression) and also in NASH-related hepatocarcinogenesis (by aberrant gene expression and mutations).

**Mutations verified in human NASH-HCCs.** Among the 80 recurrently mutated genes identified in mouse NASH-HCCs, 66 have human orthologs (Supplementary Data 2). We investigated the exome-sequencing data to examine for the presence of these mutations in 37 human NASH-HCCs (the 17 in-house samples as mentioned earlier, and 20 from The Cancer Genome Atlas (TCGA) database). We found 38 of the 66 genes were mutated in human NASH-HCCs, including 5 calcium signaling genes (Supplementary Data 4). Of note, 21 genes were recurrently mutated in at least two human NASH-HCCs (Fig. 7a). These included *MTOR* (insulin signaling, and mutational cancer driver in kidney and lung)[13], *SDK1* (cell adhesion molecule and potential driver of asbestos-associated cancers)[14], and three calcium signaling genes (*RYR1, RYR2* and *CACNA1H*).

**Mutations in calcium signaling pathway in human NASH-HCCs.** As already noted, calcium signaling genes were most

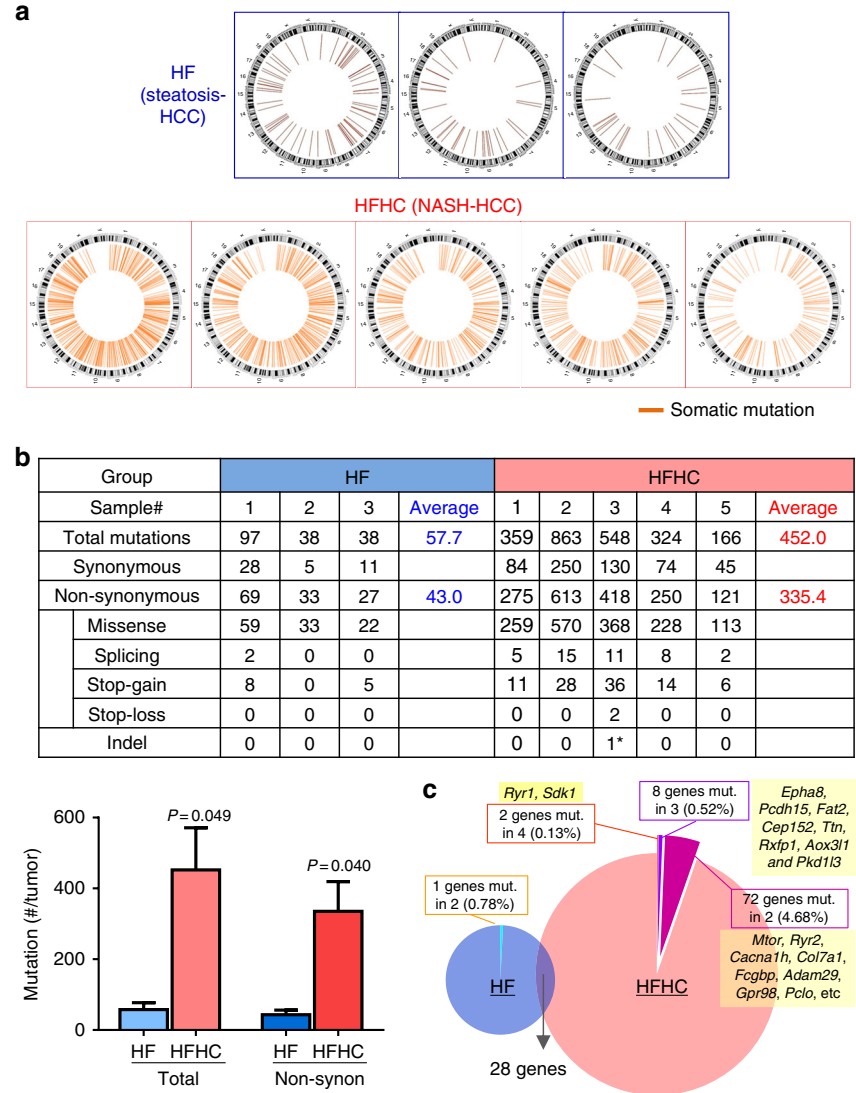

**Fig. 5** NASH-HCCs from HFHC-fed mice harbored significantly more mutations than steatosis-HCCs from HF-fed mice. **a** Circos indication of somatic mutations identified by whole-exome-sequencing. **b** Numbers of somatic mutations in each sample classified according to mutation types. *only 1 indel was found in one NASH-HCC and thus was counted in non-synonymous mutations. Comparison of the total and non-synonymous mutation numbers showed significant difference between HF and HFHC groups (unpaired *t* tests). **c** Distribution of genes recurrently mutated (in ≥2 samples) in each group and commonly mutated in both groups (non-synonymous). Numbers and percentages of genes recurrently mutated are shown in each group. HF, high-fat diet; HFHC, high-fat high-cholesterol diet

reproducibly mutated in murine NASH-HCCs, prompting us to further investigate them in human NASH-HCCs. Of note, we identified 19 genes in calcium signaling to be recurrently mutated in at least two of the 37 human NASH-HCCs. Similar to murine NASH-HCCs, most of these genes mutated in human NASH-HCCs (11/19) encode calcium channel proteins (*RYR1*, *RYR2*, *CACNA1B*, *CACNA1E*, *CACNA1H*, *CACNA1I*, *GRIN2C*, *ATP2A2*, *ATP2B4*, *SLC8A1* and *ITPR3*) (Fig. 7b). Eight genes showed significantly higher mutation frequencies in NASH-HCCs (the combined cohort (n = 37) or TCGA cohort (n = 20)) as compared to other HCCs not complicating NASH/NAFLD (TCGA, n = 353), and two of them encode calcium channel proteins (*RYR1* and *CACNA1H*) (Fig. 7c). *RYR1* was mutated most frequently in both murine (4/5 = 80%) and human (5/37 = 13.5%) HCCs associated with NASH pathology, and *RYR1* was less often mutated in other HCCs (7.37%, n = 353, TCGA data) (Fig. 7c). The 9 *RYR1* mutations identified in mouse and human NASH-HCCs were located across the whole gene, with 3 stop-

gain or truncating mutations inferring likely loss-of-function mutations (Fig. 7d). These findings suggest that genetic disruptions of calcium channels may alter calcium homeostasis in a way that contributes to NASH-related carcinogenesis.

## Discussion

Hepatic free cholesterol has been implicated in hepatic lipotoxicity[25], and may thereby activate inflammatory recruitment to fatty livers of HFHC-fed mice to induce NASH. In this study, aberrant gene expression affecting inflammatory signaling pathways (macrophage markers, cytokine, chemokine, interleukin, TGF-β and Wnt-related signaling) was identified in HFHC-associated NASH compared with HF-associated steatosis. This is consistent with a previously report that aberrant gene expression was involved in abnormal inflammation during NAFLD progression[26]. Although others have found that genes involved in Wnt signaling may be downregulated in NASH[27], our study showed upregulated expression of Wnt signaling

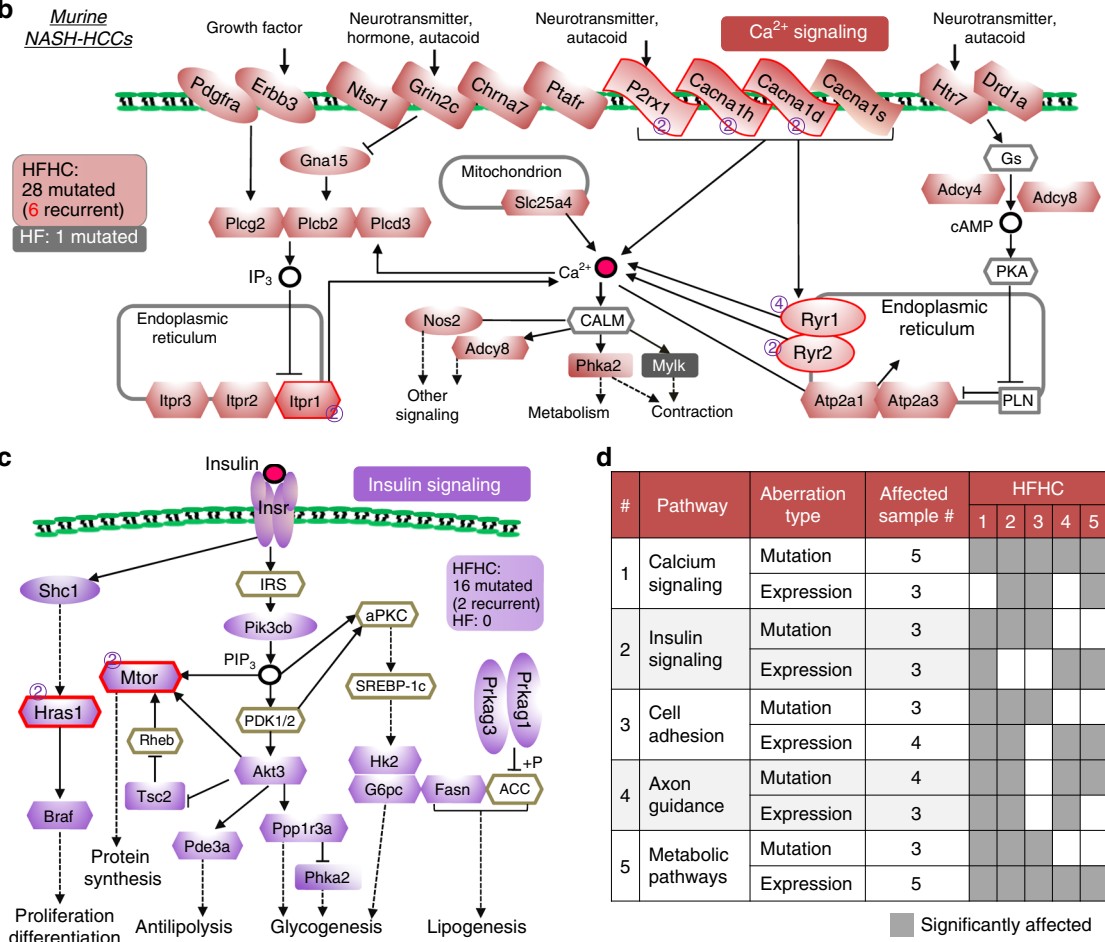

**a**

| # | Pathway | Affected HFHC tumor # | Mutated gene HFHC | Mutated gene HF | Recurrently mut.gene HFHC | Recurrently mut.gene HF | Gene# in pathway | Mutated genes in HFHC (NASH-HCCs) |
|---|---------|------------------------|-------------------|-----------------|---------------------------|-------------------------|------------------|-----------------------------------|
| 1 | Calcium signaling | 5 | 28 | 1 | 6 | 0 | 178 | *Ryr1, Ryr2, Cacna1h, Cacna1d, Itpr1, P2rx1*, etc |
| 2 | Insulin signaling | 3 | 16 | 0 | 2 | 0 | 137 | *Mtor, Hras1, Akt3, Braf, Insr, Shc1, Pik3cb*, etc |
| 3 | Cell adhesion | 3 | 12 | 0 | 2 | 0 | 149 | *Vcan, Jam2, Cntnap2, Itga6*, etc |
| 4 | Axon guidance | 4 | 17 | 0 | 2 | 0 | 131 | *Epha8, Hras1, Limk2, Efna3*, etc |
| 5 | ABC transporters | 3 | 10 | 0 | 2 | 0 | 45 | *Abcb1b, Abcc2, Abcc9, Abcb4, Cftr*, etc |
| 6 | Tight junction | 3 | 17 | 1 | 2 | 0 | 137 | *Hras1, Jam2, Actn2, Ctnna3, Exoc3*, etc |
| 7 | Metabolic pathways | 3 | 75 | 3 | 2 | 0 | 1184 | *Aox3l1, Cyp2b19, Extl1, Prps1l3, Tm7sf2*, etc |

**d**

| # | Pathway | Aberration type | Affected sample # | HFHC 1 | 2 | 3 | 4 | 5 |
|---|---------|-----------------|-------------------|--------|---|---|---|---|
| 1 | Calcium signaling | Mutation | 5 | | | | | |
| | | Expression | 3 | | | | | |
| 2 | Insulin signaling | Mutation | 3 | | | | | |
| | | Expression | 3 | | | | | |
| 3 | Cell adhesion | Mutation | 3 | | | | | |
| | | Expression | 4 | | | | | |
| 4 | Axon guidance | Mutation | 4 | | | | | |
| | | Expression | 3 | | | | | |
| 5 | Metabolic pathways | Mutation | 3 | | | | | |
| | | Expression | 5 | | | | | |

■ Significantly affected

**Fig. 6** Pathways dysregulated by genetic alterations. **a** Seven pathways were significantly dysregulated in three or more NASH-HCCs from HFHC-fed mice ($\geq 3$ genes involved in each sample, adjusted $P < 0.05$). **b** Mutated genes in calcium signaling pathway are indicated. 28 genes were mutated in NASH-HCCs, 6 of which were recurrently mutated. Only one gene in calcium signaling pathway was mutated in one of the steatosis-HCCs. Color-filled icons show mutated genes. Recurrently mutated genes were highlighted with affected NASH-HCC numbers denoted. **c** Mutated genes in insulin signaling are indicated. 16 gene were mutated in NASH-HCCs, 2 of which were recurrently mutated, while no gene was mutated in steatosis-HCCs. **d** Five pathways were enriched commonly by mutations and aberrant gene expression in NASH-HCCs

genes in NASH compared to steatosis, including *Ctnnb1* and *Myc* (Fig. 2). As similar upregulated expression of Wnt signaling genes has been reported in HCC, our finding supports the hypothesis that NASH is more prone to malignant transformation than steatosis through inducing inflammatory pathways, including those pertinent to Wnt signaling.

One of our important and novel findings is that the aberrantly expressed genes and pathways accelerated DEN-induced HCC

based on NASH. We further confirmed that some of these genes were also aberrantly expressed in human NASH-associated HCCs as compared to their adjacent non-tumorous liver. We demonstrated for the first time the aberrant expression of cancer-related genes (*ALCAM, ITGA6, DDIT3, MAP3K6* and *PAK1*) and metabolism-related genes (*ALDH18A1, CAD, CHKA, POLD4, PSPH, SQLE* and *CFD*) in human NASH-HCCs. These findings highlight their particular importance in the development of

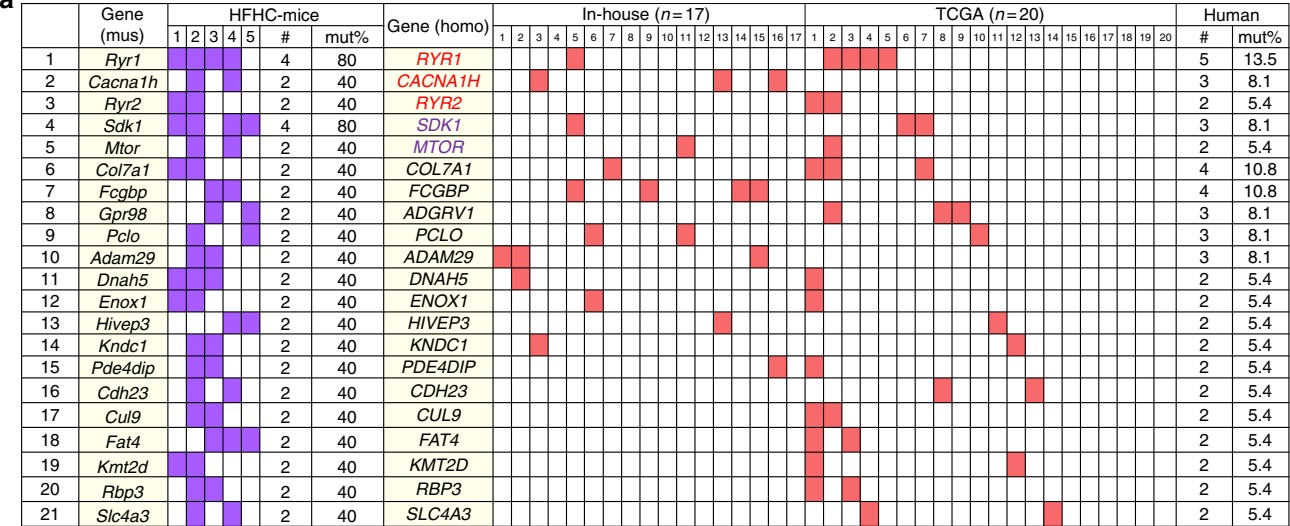

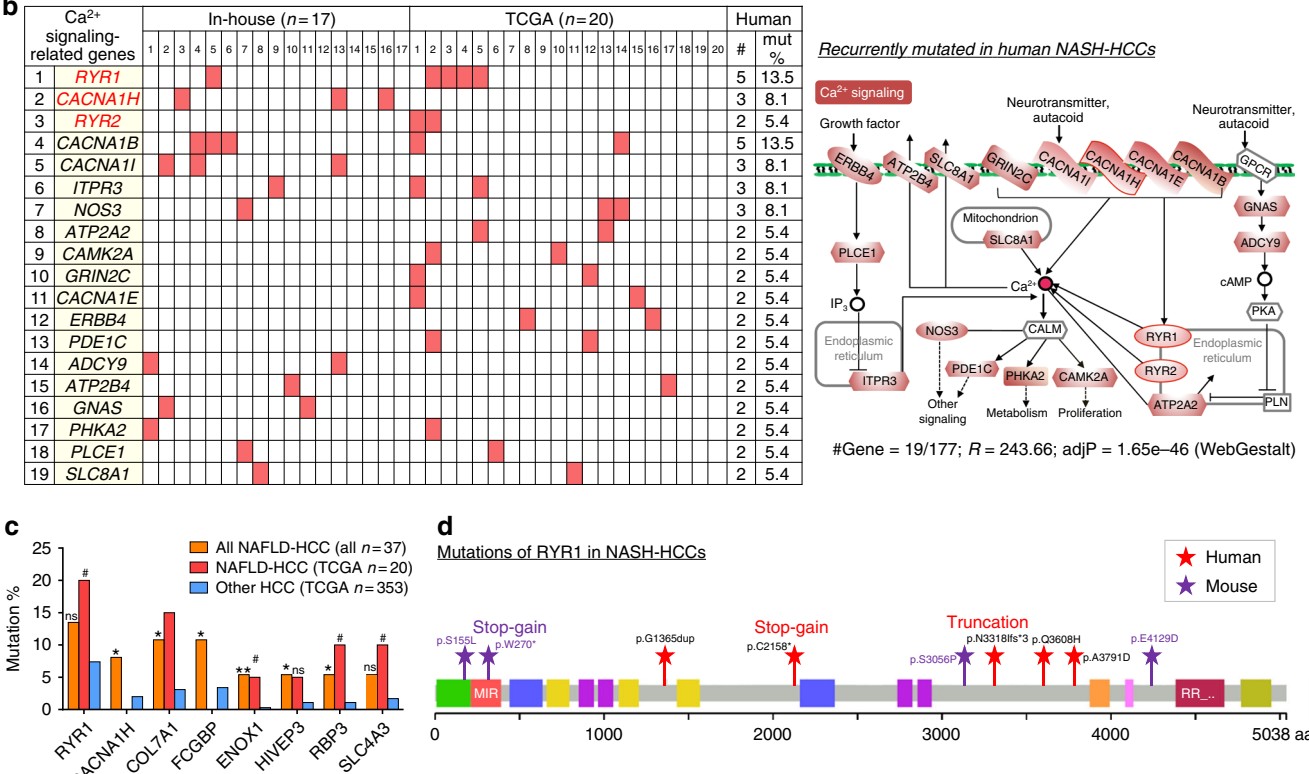

**Fig. 7** Validation of recurrently mutated genes in human NASH-HCCs. **a** Recurrently mutated genes identified in mouse NAHS-HCCs were checked in 37 human NASH-HCCs (17 in-house and 20 from TCGA). Recurrent mutations in 21 genes were verified in human NASH-HCCs. **b** Recurrently mutated calcium signaling-related genes in human NASH-HCCs. **c** Mutation frequencies of eight genes were higher in HCCs with risk history of NAFLD than others without NAFLD (Chi-square test). *$P < 0.05$ and **$P < 0.001$ for all NAFLD-HCCs (in-house and TCGA, $n = 37$) versus other HCCs; #$P < 0.05$ for NAFLD-HCCs from TCGA ($n = 20$) vs. other HCCs; ns non-significant as compared with other HCCs. **d** Schematic illustration of the mutation sites in *Ryr1* identified in mouse and human NASH-HCCs

NASH-related HCC. Tumor-promoting roles of MAP3K6, PAK1, ALCAM and ITGA6 have been reported[28–31], thus their upregulation may also function in the development of NASH-related HCC. DDIT3 is a multifunctional transcription factor in endoplasmic reticulum stress and is associated with inflammatory responses, thus upregulated DDIT3 should be involved in the development of NASH-HCC. Upregulation of the metabolic genes (*ALDH18A1, CAD, CHKA, POLD4, PSPH, SQLE*) may function in accelerating cellular metabolism. Of note, SQLE

(squalene epoxidase) encodes a rate-limiting enzyme in cholesterol biosynthesis and exerts oncogenic roles in cancers[32–34]. Our team has recently revealed that upregulation of SQLE plays an important oncogenic role in driving NAFLD-HCC[32]. *CFD* encodes human adipsin that functions as an adipokine and regulates insulin secretion[35]. Downregulation of *CFD* expression may be associated with dysregulation of insulin signaling in NASH-HCC. Collectively, dietary cholesterol induced hepatic aberrant expression of genes and their associated necro-

inflammatory (TGF-β signaling, chemokine signaling, etc), oncogenic signaling pathways (MAPK signaling, axon guidance and cell adhesion, etc) and metabolic pathways to contribute to the development of NASH-HCC.

In contrast to liver cancers in HFHC-fed mice, significantly fewer genes were mutated during hepatocarcinogenesis in HF-fed mice. It therefore seems likely that alterations of gene expression contribute more importantly to hepatocarcinogenesis than do gene mutations in HF-fed mice. In addition to the anticipated dysregulated expression of cancer-related and metabolic pathways, 13 out of the 16 genes related to cholesterol synthesis were significantly upregulated (all $P < 0.05$; Supplementary Fig. 5). We interpret this phenomenon as indicating that without an augmented level of cholesterol intake, hepatocytes and/or HCC precursor cells are required to undergo de novo cholesterol synthesis to promote cancer development. This concept supports a previous hypothesis that cancer cells need excess cholesterol and intermediates of the cholesterol biosynthesis pathway to maintain cell proliferation, presumably because a large quantum of cholesterol is required for the synthesis of cell membrane[36]. High levels of dietary cholesterol could therefore accelerate hepatocarcinogenesis by bypassing the requirement for enhanced endogenous cholesterol synthesis. Recent data have shown that major cholesterol metabolites, such as 27-hydroxycholesterol and 6-oxocholestan-3beta,5alpha-diol, display tumor-promoter properties[37,38]. Therefore, we further analyzed the expression of a curated list of 63 oxysterol pathway genes in mus musculus (Supplementary Data 5) known to be involved in the main oxysterol pathway, oxysterol metabolism, oxysterol pathway genes regulation and oxysterol transport[39]. We observed 47 (out of 63) dysregulated oxysterol pathway genes in dietary cholesterol-associated NASH-livers compared to non-cholesterol-associated steatosis-livers, as well as 33 dysregulated oxysterol pathway genes in cholesterol-associated NASH-HCCs compared to their adjacent non-tumor livers (Supplementary Fig. 6). We interpret these findings as supporting the proposal that dysregulated oxysterol metabolism is involved in the development of cholesterol-associated NASH, and also in accelerated hepatocarcinogenesis associated with NASH.

Exome-sequencing analysis showed that cholesterol-induced NASH-HCCs in mice harbored up to 7.8-fold more mutations compared to steatosis-HCCs. Oxidative DNA damage has been implicated in hepatocarcinogenesis in human NASH[40], and inflammation has also been associated with DNA lesioning. Therefore, mutational difference in HCCs between HFHC- and HF-fed mice might be attributable to cholesterol-associated NASH development involving more severe inflammation. The increased number of mutations in HFHC tumors may also be partially due to more advanced stages of HCC or multiclonality as spatial genetic divergence has been observed in HCC[41].

Due to the small sample size of human NASH-HCC data available, it is currently not feasible to use stringent methods such as MutSigCV, which takes into account of background mutations and other confounding covariates to identify significantly mutated genes. Our study took the advantage of cross-species oncogenomics to identify genetic alterations in mouse models and then validated such changes in human samples so as to identify the potential key genes associated with human NASH-HCC. We identified 21 recurrently mutated genes in mouse NASH-HCC models and further validated in human NASH-HCC. Key replicated genes included the mutational cancer drivers *MTOR* and *SDK1*, and three calcium signaling-related genes *RYR1*, *CACNA1H* and *RYR2*. Of note, *MTOR* has been determined to be an HCC driver gene using MutSigCV[42], and was identified to be one of the most important genes associated with NASH-HCC in this study, suggesting the special significance of *MTOR* mutations in this subtype of HCC. Whether other genes

identified in this study are true driver genes in NASH-HCC need further validation in a larger cohort of samples as well as functional investigation. One exome analysis of 10 human NAFLD-HCC patients identified two previously unreported somatic mutations (*FGA* and *SYNE1*) that might contribute to NAFLD-HCC, as well as other mutations in known HCC-associated genes such as *TERT* promoter, *CTNNB1*, *TP53*, *ARID1A*, *ARID2*, *TSC2*, *ACVR2A*, *NFE2L2* and *AXIN1*[43]. Conversely, another study failed to find *CTNNB1*, *TERT* and *TP53* mutations but instead noted frequent expressional activation of IL-6/JAK/STAT in steatohepatitic HCCs as compared with non-steatohepatitic HCCs[44]. Taken together, we interpret the present data as indicating that dietary cholesterol, either directly or via the liver pathology of NASH, induces recurrent mutations and their associated oncogenic pathways in a way that contributes to accelerated onset and spread of carcinogen-induced HCC.

Integrated analysis of gene expression and mutations revealed that mutations exerted little influence on the levels of gene expression. Nevertheless, aberrant expression and mutations in NASH-HCCs affected different genes that were commonly enriched in five core pathways of calcium signaling, insulin signaling, cell adhesion, axon guidance and metabolism. Pertinent to their importance in the development of NASH-HCC, these pathways were already dysregulated in NASH liver by aberrant gene expression. This presumably reflects the involvement of progressive changes of these pathways during NASH-related liver carcinogenesis. Dysregulation of such pathways has been reported in other cancers[45–49].

In the present study, the calcium signaling gene *RYR1* that was mutated most frequently in both murine and human NASH-HCCs appears to be of pathogenic significance. Calcium signaling plays an important physiological role in metabolism, and its dysregulation has previously been implicated in cancer development[23,24]. Furthermore, calcium signaling is known to be affected by cellular cholesterol content[21,22]. Intracellular calcium ions are important second messengers that regulate gene transcription, cell proliferation, migration and death. Thus, altered intracellular calcium homeostasis is involved in tumor initiation, angiogenesis, tumor progression and metastasis. Some important calcium channels, transporters and calcium-ATPases are altered in human cancer patients[50]. It is therefore salient to note that most recurrently mutated calcium signaling genes identified in NASH-HCCs (all 6 in mice and 11/19 in human) encode key calcium channels; genetic disruptions of these channels may alter calcium homeostasis and consequently contribute to carcinogenesis. These findings highlight the potential importance of the dysregulated calcium signaling during hepatocarcinogenesis in both experimental and clinical NASH. Targeting derailed calcium signaling by blockers, inhibitors or regulators for calcium channels/transporters/pumps in cancer therapy has become an emerging research area[50]. The altered calcium signaling-related genes identified in this study with available compounds for targeted therapy include RYR (encoded by *RYR1* and *RYR2*) and IP3R3 (encoded by *ITPR3*) channels, for which chemotherapeutic agents include RYR agonists (4-chloro-m-cresol, caffeine and its analogs) and IP3R3 inhibitors (heparin and caffeine). Notably, intake of two or more cups of coffee each day reduces HCC incidence by more than 30% in patients with cirrhosis[51]. Future study to investigate the efficacy of these compounds targeting RYR and IP3R channels in inhibiting NASH-HCC is warranted.

In summary, this study provides the first systematic overview of altered gene expression, gene mutations and the affected signaling networks in HCC developing in the context of NASH compared to steatosis using appropriate murine models (Fig. 8). Key findings are that dietary cholesterol causes NASH via dysregulating expression of genes involved in inflammation and

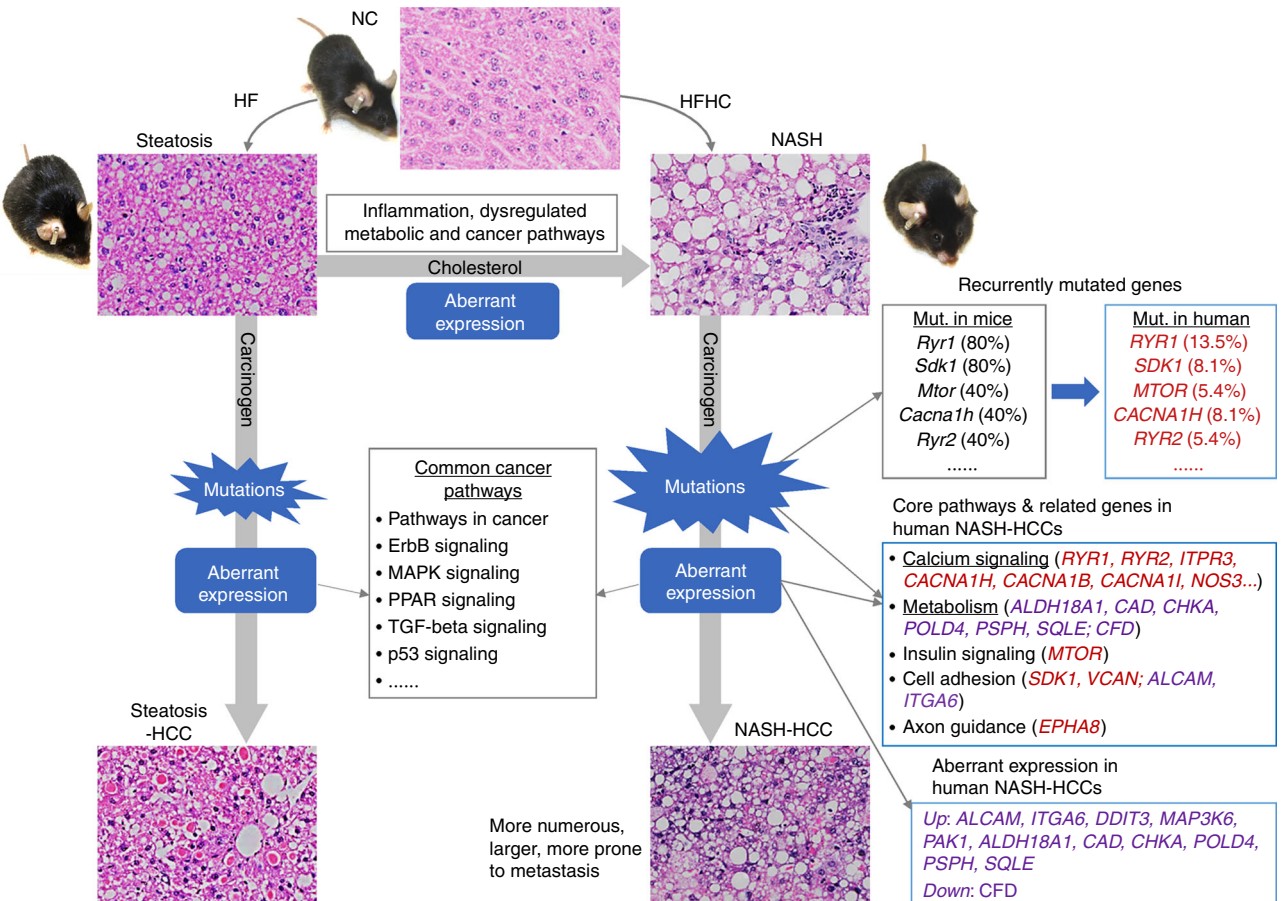

**Fig. 8** Schematic summary of this study. NASH development promoted by dietary cholesterol is associated with aberrant gene expression linked with activated inflammatory signaling, dysregulated metabolic and cancer-related pathways. Dietary cholesterol-induced NASH-HCC development in mice is associated with aberrant gene expression, genomic mutations and the associated core pathways. The identified aberrant gene expression and mutations were verified in human NASH-HCCs. NC, normal chow; HF, high-fat diet; HFHC, high-fat high-cholesterol diet

metabolism, and augments the accelerated development of HCC through induction of oncogenic mutations and gene expression. In particular, we identified novel aberrant gene expression, mutations and core oncogenic pathways, most notably those linked to calcium transport channels that contribute to HCC with cholesterol-induced NASH, and demonstrated analogous changes in clinical samples. Further studies should interrogate the functional consequences of such molecular changes and test whether they can be exploited pharmacologically as chemoprevention or therapy of NASH-related HCC.

## Methods

**Animals and diets**. From 6 weeks of age, male C57BL/6 J mice were randomly fed one of three diets (Specialty Feeds, Glenn Forrest, Western Australia) for 26 weeks: 1) normal chow (n = 9), 2) high saturated-fat ([HF], 23% fat, 45% carbohydrate, 20% protein, 0% cholesterol w/w) (n = 10), and 3) combined high saturated-fat and high cholesterol ([HFHC], 23% fat, 45% carbohydrate, 20% protein, 0.19% cholesterol w/w) (n = 10). To accelerate HCC development in mice, a single injection of chemical carcinogen DEN (10 mg/kg, Sigma-Aldrich, St. Louis, MO) intraperitoneally was applied at age 15 days (0.9% saline to controls) to induce HCC. This is the most widely used chemical to induce liver cancer in mice[52–54]. At 32 weeks of age, mice were euthanized, and serum and liver were collected for analyses. Tumors and non-tumorous liver tissues were fixed in 10% neutral-buffered formalin or snap-frozen in liquid nitrogen. All animal experiments adhered to protocols approved by Australian National University's Animal Experimentation Ethics Committee.

**Assessment of metabolic phenotypes and liver histology**. Blood glucose and lipids were measured using automated techniques for clinical pathology at the Canberra Hospital. To evaluate insulin sensitivity, we performed intraperitoneal glucose tolerance tests (IPGTT) 2 weeks prior to sacrifice. Hepatic triglyceride, free

cholesterol, cholesterol ester fractions were quantified using high-performance liquid chromatography and results normalized to liver wet weight (g), as described previously[8]. Liver sections (5 μm) were stained with hematoxylin and eosin and assessed blind by an expert liver pathologist (Matthew M Yeh) according to the NAFLD activity score[55].

**Whole-exome sequencing**. DNA samples of paired tumor and adjacent non-tumorous livers of 3 HF-fed mice and 5 HFHC-fed mice were subjected to whole-exome sequencing. Exome capture was performed using the SureSelect Target Enrichment System (Agilent, Santa Clara, CA). Captured libraries were sequenced by HiSeq2000 (Illumina, San Diego, CA). Sequencing mean depths were > 100-fold and coverages of the targeted exome were > 99% for all samples (Supplementary Table 3).

**Mouse genome alignment**. The whole-exome sequencing reads were mapped to the UCSC mm10 reference genome (http://genome.ucsc.edu/) using the Burrows-Wheeler Aligner[56]. The likely PCR duplications with the same match interval on the genomic sequence duplications were removed by Samtools[57]. Local realignment around small insertions and deletions (Indels) were performed using GATK to improve alignment quality and help reduce the false discovery rate in the following mutation detection analysis[58].

**Single-nucleotide variants (SNVs) calling**. Candidate SNVs with mutated allele frequencies > 15% in cancer tissue and no more than 2% in paired normal tissue (Fisher's exact test, $P < 0.05$) were filtered by the following thresholds: high-quality ($Q > 10$) coverage ≥ 10X for each sample; reads supporting the mutated allele not a result of sequencing error (Binomial test, $f = 0.1$, $P > 0.01$); sequencing quality scores for mutated alleles not lower than normal alleles (Wilcoxon rank sum test, $P > 0.01$); mutated alleles not come from repeatedly aligned reads (Fisher's exact test, $P > 0.01$). Mutant alleles should not be enriched in 10 bps of 5' or 3' ends of reads (Fisher's exact test, $P > 0.01$). Annotation was performed by ANNOVAR[59].

**Indels calling**. Candidate somatic indels were firstly predicted by GATK SomaticIndelDetector with default parameters[58]. Predicted indels were then filtered if (1) total coverage at the site < 30×; (2) average mapping qualities of consensus indel-supporting reads < 30; (3) average number of mismatches per consensus indel-supporting read ≥ 2; (4) average quality of bases from consensus indel-supporting reads < 20; (5) median of indel offsets from the ends/starts of the reads within 10 bp, or (6) percentage of forward- or reverse-aligned indel-supporting reads < 20%. Resulting somatic indels were filtered against dbSNP128. Annotation was performed by ANNOVAR[59].

**Identification of known or potential cancer driver genes**. Mutated genes identified in mouse models were cross-checked using the Catalog of Cancer Driver Mutations[60] (version 2016.5; https://www.intogen.org/analysis/home).

**Genome-wide expression analysis using microarray**. Gene expression profiles of tumors and adjacent non-tumor liver tissues from 3 HF- and 5 HFHC-fed mice (same samples as for exome sequencing) were analyzed using the SurePrint G3 Mouse GE 8 × 60 K Microarray Kit (Agilent Technologies, Palo Alto, CA), which contained 39,430 Entrez Gene RNAs. In brief, RNA was extracted using Qiazol reagent (Qiagen, Valencia, CA). The cDNA probes were prepared from 5 μg of total RNA labeled with Cy5-dUTP (red) or Cy3-dUTP (green) by reverse transcription (Amersham Biosciences, Piscataway, NJ). Two labeled cDNAs were competitively hybridized to the microarray. Signal intensities were analyzed using a SureScan microarray scanner (Agilent Technologies). Array data were presented as log base 2 ratio of the Cy5/Cy3 signals. Gene expression patterns between HF and HFHC groups were compared using unsupervised hierarchical clustering.

**Pathway enrichment analyses**. Mutated genes or differentially expressed genes were subjected to KEGG pathway enrichment analysis using Gene Set Analysis Toolkit V2 (http://bioinfo.vanderbilt.edu/webgestalt/)[61,62]. The hypergeometric test statistical method and the BH multiple test adjustment method were used. All mouse genes were used as reference. Pathways with at least 4 aberrantly expressed genes/3 mutated genes and adjusted $P < 0.05$ were considered as significantly enriched. WikiPathways analysis was also used for pathway identification and data visualization (http://www.wikipathways.org)[63].

**Human samples, gene expression and mutation analyses**. Recurrently mutated genes identified in tumors of HFHC-fed mice were further investigated in human NASH-HCC samples. Tumors and paired non-tumorous liver tissues from 17 patients diagnosed with HCC complicating NASH were obtained from Prince of Wales Hospital, The Chinese University of Hong Kong and Zhongshan Hospital of Fudan University. All patients gave informed consent for participation in this study. This study was approved by the ethics committee of the Chinese University of Hong Kong and the clinical research ethics committee of Zhongshan Hospital of Fudan University. Tumor and paired non-tumorous liver tissues were subjected to RNA sequencing and whole-exome sequencing. Gene expression was analyzed and compared between tumors and adjacent non-tumorous liver as our previous description[47]. Somatic mutations were called as our previous description[64]. Expression and somatic mutations in the genes of interest were then checked in the 17 cases. RT-qPCR was performed to verify the expression of selected genes in 12 cases, with primers listed in Supplementary Table 4.

To further increase the sample size, we also checked mutations in the genes of interest in the 20 human HCCs associated with NAFLD that had been analyzed by exome sequencing in TCGA project. Mutation data were downloaded from TCGA database at cBioPortal Cancer Genomics (http://www.cbioportal.org/public-portal/index.do).

**Statistical analyses**. All measurements are shown as means ± SD or means ± SEM where appropriate. ANOVA corrected for multiple comparisons using Tukey's test was used to compare data of each two groups when multiple groups were involved, and multiplicity adjusted $P$ values were reported for each comparison. Unpaired $t$ tests were used for comparison of data involving only HF- and HFHC-fed groups. Paired $t$ tests were used for comparison of expression levels between paired tumor and adjacent non-tumorous livers from mice and human. FDR was adjusted using the Benjamini–Hochberg method for RNA seq data. Mutation frequencies were compared by chi-square test. All statistical tests, except FDR, were performed using Graphpad Prism 5.0 (Graphpad Software Inc., San Diego, CA), and a two-tailed $P$ value of < 0.05 was considered statistically significant.

## Data availability

Exome sequencing data are deposited in NCBI Sequence Read Archive (http://www.ncbi.nlm.nih.gov/sra) under the accession number SRP115597 (https://www.ncbi.nlm.nih.gov/Traces/study/?acc = PRJNA398450). Microarray data are deposited in EMBL-EBI ArrayExpress (https://www.ebi.ac.uk/arrayexpress/) under accession number E-MTAB-6002 (https://www.ebi.ac.uk/arrayexpress/experiments/E-MTAB-6002/).

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

## Acknowledgements

The results shown here are in part based upon data generated by the TCGA Research Network: http://cancergenome.nih.gov/. Initial dietary experiments leading up to the conduct of these studies were planned by Dr Claire Larter, and Vanessa Barn who provided excellent technical support in the animal work. This project was supported by the RGC-GRF Hong Kong (14106415, 14111216, 14163817); Collaborative Research Fund RGC (Ref. No. C4041-17GF); Vice-Chancellor's Discretionary Fund CUHK; CUHK Direct grant for research (4054268); Shenzhen Virtual University Park Support Scheme to CUHK Shenzhen Research Institute; Australian National Health and Medical Research Council (NHMRC) project grants 585411, 1084136, 1044288 and 1120898.

## Author contribution

J.Q.L. conceived and performed the analyses, summarized all the data and figures, and wrote the manuscript. J.Y. and G.F. conceived and supervised the study and revised the manuscript. J.Y., N.T. and G.F. secured funding. N.T., S.P., L.X. E.S.H.C, J.C., E.A., W.G. H. and M.M.Y. performed experiments. X.L. helped bioinformatics analysis of mutation calling. L.D. and J.Y. provided human samples. G.N.I., G.F. and J.J.Y.S. provided expertise and feedback.

## Additional information

**Competing interests:** The authors declare no competing interests.

