## [Peer Review File · Nature Communications]

Reviewers' comments:

Reviewer #1 (Remarks to the Author):

The authors studied the impact of dietary cholesterol on hepatocellular carcinoma (HCC). They compared the impact of dietary cholesterol on diethylnitrosamine(DEN)-injected mice fed with high fat diets with or without high cholesterol. They used mRNA microarray and whole exome sequencing analyses to identify genetic aberrations and specific variation in gene expression and validated molecular changes in 37 human NASH-HCC.

This study is interesting but mainly descriptive and suffered from approximations which make the reader confused regarding the models and the fact that models get treatment with DEN. Variation in calcium pathways as a consequence in cholesterol enriched diet is an interesting observation but the authors didn't give molecular insights on the origin of this deregulation neither they proposed therapeutic strategies.

Importantly the authors did not integrate recent data from the literature related to cancer promotion by cholesterol and cancer. In particular recent data highlighted that 27hydroxycholesterol and 6-oxocholestan-3beta,5alpha-diol displayed tumor promoter properties either by acting on the tumor microenvironment or directly on the tumor. This requires that the authors performed analyses of the oxysterol profile in the blood and in the tumors of mice as well. They must study these specific these metabolic pathways including the effectors of these oxysterols. In my opinion the stheshold of 2 they have chosen for transcription analyses is too high and they probably missed oncogenic drivers especially if they are enzymes that produce oncometabolites. An increase of 1.2 is enough to activate a whole metabolic pathway. This observation could apply to nuclear receptors for which no correlation exists between mRNA and protein expression.

Reviewer #2 (Remarks to the Author):

The authors use a mouse model where they can specifically examine the impact of cholesterol on non-alcoholic fatty liver disease and HCC formation. They compare their results with human tumors. Overall, the authors do highlight some of their novel findings, but could do more in the discussion to make it clear how the work expands upon previous studies in human non-alcoholic fatty liver disease and HCC. In addition, there are some concerns about statistical analyses employed.

Major comments:

- Simple t tests are not appropriate in situations where there are more than 2 groups. This occurs frequently in Figure 1.
- In figure 1 is there a reason why control diet treated mice were not included in d, e, f1 and f2?
- Be very careful with wording so that results are not over-stated. For example it is stated "Dietary cholesterol causes NASH through up-regulating inflammatory genes" yet there is no functional proof that the up-regulation of inflammatory genes is responsible for the phenotype.
- For the mouse tumor genome sequencing, the HFHC tumors were "multiple large" while most HF tumors are "single small". Is it possible that the large tumors are actually multi-clonal (e.g. small tumors growing close together) and that could influence the number of mutations present?
- Were steatosis-HCC and NASH-HCC gene expression ever compared directly to each other? Would that be a more valid comparison than that which is presented at the top of page 11?
- Relating to the human RNA seq data presented in figure 4, it is not usual to see RNA seq data

filtered for a small set of genes but instead to analyze the entire transcriptome. I am concerned that the statistics employed (simple paired t-tests) do not account for multiple hypothesis testing. If the data is being presented to simply verify data from mice, then it would seem like qPCRs would have been more powerful.

- Please provide a description of the statistical analysis performed in figure 7c.

- Although the authors do highlight some of the novel findings of this study in the discussion, this reviewer feels like even more should be done to discuss the novel findings, similar findings, and findings that are not in agreement from previous related studies. For example, there is at least one study I found looking at gene expression in human steatosis and NASH that don't seem to be discussed and neither is the actual TCGA publication from 2017.

Minor points:

- In methods for liver sections by 5 m do you mean 5 micron?

- In Figure 5a, "HFHC" is cropped

- Page 11 lines 19 and 20 why were the genes of "particular interest"? Also, the use of "etc" in gene lists could be somewhat confusing if a reader happened to think that was a gene symbol.

- Please provide scale bars on images in 1f1.

- For lung mets can you include a supplementary figure?

Response to the comments of Reviewers in relation to the manuscript:

MS# NCOMMS-17-31327-T, “Dietary cholesterol promotes steatohepatitis-related hepatocellular carcinoma by inducing aberrant gene expression in metabolism and mutations in calcium signaling”

COMMENTS are written in *italics* and RESPONSES in normal text.

Reviewer #1 (Remarks to the Author):

The authors studied the impact of dietary cholesterol on hepatocellular carcinoma (HCC). They compared the impact of dietary cholesterol on diethylnitrosamine (DEN)-injected mice fed with high fat diets with or without high cholesterol. They used mRNA microarray and whole exome sequencing analyses to identify genetic aberrations and specific variation in gene expression and validated molecular changes in 37 human NASH-HCC.

This study is interesting but mainly descriptive and suffered from approximations which make the reader confused regarding the models and the fact that models get treatment with DEN. Variation in calcium pathways as a consequence in cholesterol enriched diet is an interesting observation but the authors didn't give molecular insights on the origin of this deregulation neither they proposed therapeutic strategies.

RESPONSE:

We appreciate the comments. We have added detailed information in the revised text regarding our study models and that DEN treatment was given in these models (**Pages 5, 9 and 15**), as well as the molecular insights on the origin of the deregulation of calcium pathway and potential therapeutic strategies (**Page 18**). Details are shown below:

Page 5, line 12: To accelerate HCC development in mice, a single injection of chemical carcinogen DEN (10 mg/kg, Sigma-Aldrich, St. Louis, MO) intraperitoneally was applied at age 15 days (0.9% saline to controls) to induce HCC. This is the most widely used chemical to induce liver cancer in mice.¹¹⁻¹³

Page 9, line 23: To explore the role of dietary cholesterol in HCC development, we compared DEN-induced hepatocarcinogenesis in mice fed normal chow, HF, or HFHC diets.

Page 15, line 14: One of our important and novel findings is the identification of aberrantly expressed genes and pathways specifically associated with development of accelerated DEN-induced HCC in the context of murine NASH.

Page 18, line 4: In the present study, the calcium signaling gene *RYR1* that was mutated most frequently in both murine and human NASH-HCCs appears to be of

pathogenic significance. Calcium signaling plays an important physiological role in metabolism, and its dysregulation has previously been implicated in cancer development.^{37, 38} Furthermore, calcium signaling is known to be affected by cellular cholesterol content.^{35, 36} Intracellular calcium ions are important second messengers that regulate gene transcription, cell proliferation, migration and death. Thus, altered intracellular calcium homeostasis is involved in tumor initiation, angiogenesis, tumor progression and metastasis. Some important calcium channels, transporters and calcium-ATPases are altered in human cancer patients.⁶² It is therefore salient to note that most recurrently mutated calcium signaling genes identified in NASH-HCCs (all 6 in mice and 11/19 in human) encode key calcium channels; genetic disruptions of these channels may alter calcium homeostasis and consequently contribute to carcinogenesis. These findings highlight the potential importance of the dysregulated calcium signaling during hepatocarcinogenesis in both experimental and clinical NASH. Targeting derailed calcium signaling by blockers, inhibitors or regulators for calcium channels/transporters/pumps in cancer therapy has become an emerging research area.⁶² The altered calcium signaling-related genes identified in this study with available compounds for targeted therapy include RYR (encoded by *RYR1* and *RYR2*) and IP3R3 (encoded by *ITPR3*) channels, for which chemotherapeutic agents include RYR agonists (4-chloro-m-cresol, caffeine and its analogs) and IP3R3 inhibitors (heparin and caffeine). Notably, intake of two or more cups of coffee each day reduces HCC incidence by more than 30% in patients with cirrhosis.⁶³ Future study to investigate the efficacy of these compounds targeting RYR and IP3R channels in inhibiting NASH-HCC is warranted.

Importantly the authors did not integrate recent data from the literature related to cancer promotion by cholesterol and cancer. In particular recent data highlighted that 27hydroxycholesterol and 6-oxocholestan-3beta,5alpha-diol displayed tumor promoter properties either by acting on the tumor microenvironment or directly on the tumor. This requires that the authors performed analyses of the oxysterol profile in the blood and in the tumors of mice as well. They must study these specific these metabolic pathways including the effectors of these oxysterols. In my opinion the stheshold of 2 they have chosen for transcription analyses is too high and they probably missed oncogenic drivers especially if they are enzymes that produce oncometabolites. An increase of 1.2 is enough to activate a whole metabolic pathway. This observation could apply to nuclear receptors for which no correlation exists between mRNA and protein expression.

RESPONSE:

Thanks for the suggestions. In the revised text, we have integrated recent data from

the literature relating cancer promotion and cholesterol including 27hydroxycholesterol⁵³ and 6-oxocholestan-3beta,5alpha-diol.⁵⁴ We intended to perform analyses of oxysterol profile in the blood and tumors of mice. As the samples from mice have been used largely for blood biochemical assays and genomic analyses, we do not have enough materials of both blood and tissue samples left from mice to perform the oxysterol profile assay. Nevertheless, we have re-analyzed the expression of oxysterol pathway genes in mice using a threshold fold-change of 1.2 as suggested. We identified 47 out of the 63 oxysterol pathway genes (Kloudova A et al, Clin Endocrinol. 2017;86(6):852-61) to be dysregulated in HFHC-associated NASH (Supplementary Fig. 6A), suggesting that dietary cholesterol may contribute to dysregulation of oxysterol metabolism in NASH development before HCC. Moreover, we have identified 33 differentially expressed oxysterol metabolism genes in HFHC-tumors compared with adjacent non-tumor tissues (Supplementary Fig. 6B), demonstrating that dysregulation of oxysterol metabolism is involved in the development of HFHC-tumors. This information has now been added to the discussion as follows:

Page 16, line 20: Recent data have shown that major cholesterol metabolites, such as 27-hydroxycholesterol and 6-oxocholestan-3beta,5alpha-diol, display tumor promoter properties.^{51, 52} Therefore, we further analyzed the expression of a curated list of 63 oxysterol pathway genes in *mus musculus* (Supplementary Table 7) known to be involved in the main oxysterol pathway, oxysterol metabolism, oxysterol pathway genes regulation and oxysterol transport.⁵³ We observed 47 (out of 63) dysregulated oxysterol pathway genes in dietary cholesterol-associated NASH-livers compared to non-cholesterol-associated steatosis-livers, as well as 33 dysregulated oxysterol pathway genes in cholesterol-associated NASH-HCCs compared to their adjacent non-tumor livers (Supplementary Fig. 6). We interpret these findings as supporting the proposal that dysregulated oxysterol metabolism is involved in the development of cholesterol-associated NASH, and also in accelerated hepatocarcinogenesis associated with NASH.

Reviewer #2 (Remarks to the Author):

The authors use a mouse model where they can specifically examine the impact of cholesterol on non-alcoholic fatty liver disease and HCC formation. They compare their results with human tumors. Overall, the authors do highlight some of their novel findings, but could do more in the discussion to make it clear how the work expands upon previous studies in human non-alcoholic fatty liver disease and HCC. In addition, there are some concerns about statistical analyses employed.

RESPONSE: Thanks for the comments. We have added more information in the discussion to make it clear how our work expands on previous studies in human non-alcoholic fatty liver disease and HCC (**Page 15, line 2 – Page 18, line 26**).

Major comments:

- *Simple t tests are not appropriate in situations where there are more than 2 groups. This occurs frequently in Figure 1.*

RESPONSE:

Simple t tests in Figure 1 have been changed to ANOVA Tukey's test for multiple comparisons, with description added in Methods (**Page 8, line 19**) and Figure Legend (**Page 26, line 14**) as shown below:

Page 8, line 19: ANOVA corrected for multiple comparisons using Tukey's test was used to compare data of each two groups when multiple groups were involved, and multiplicity adjusted P values were reported for each comparison. Unpaired t tests were used for comparison of data involving only HF- and HFHC-fed groups.

Page 26, line 14: Data in b-g between each two groups were compared using ANOVA turkey's multiple comparison tests.

- *In figure 1 is there a reason why control diet treated mice were not included in d, e, f1 and f2?*

RESPONSE:

In the revised Figure 1, control diet treated mice have now been included in d, e, f1 and f2.

- *Be very careful with wording so that results are not over-stated. For example it is stated "Dietary cholesterol causes NASH through up-regulating inflammatory genes" yet there is no functional proof that the up-regulation of inflammatory genes is responsible for the phenotype.*

RESPONSE:

Thank you for the advice. We have revised the statements accordingly.

Page 10, line 2: Dietary cholesterol up-regulated inflammatory genes during NASH development.

Page 10, line 14: These findings demonstrate the necro-inflammatory nature of gene expression changes in liver in response to HFHC diet that might be associated with NASH and with accelerated development of HCC.

- *For the mouse tumor genome sequencing, the HFHC tumors were "multiple large" while most HF tumors are "single small". Is it possible that the large tumors are actually*

multi-clonal (e.g. small tumors growing close together) and that could influence the number of mutations present?

RESPONSE:

Yes it is possible that the HFHC tumors subjected to sequencing were multi-clonal. We have added this possibility in the discussion as following:

Page 17, line 4: Oxidative DNA damage has been implicated in hepatocarcinogenesis in human NASH,⁵⁴ and inflammation has also been associated with DNA lesioning. Therefore, mutational difference in HCCs between HFHC- and HF-fed mice might be attributable to cholesterol-associated NASH development involving more severe inflammation. The increased number of mutations in HFHC tumors may also be partially due to multiclonality as spatial genetic divergence has been observed in HCC.⁵⁵

- Were steatosis-HCC and NASH-HCC gene expression ever compared directly to each other? Would that be a more valid comparison than that which is presented at the top of page 11?

RESPONSE:

Thanks for the suggestions. We have compared steatosis-HCC and NASH-HCC gene expression directly, and found that 4,660 gene (2-fold) were differentially expressed in NASH-HCC as compared to steatosis-HCC. This information has now been included in Result section as shown below:

Page 10, line 20: To understand the molecular basis for accentuated hepatocarcinogenesis in HFHC-fed versus HF-fed mice, we compared gene expression profiles of NASH-HCC with steatosis-HCC directly. We noted that 4,660 genes were aberrantly expressed (2-fold or more) in NASH-HCC versus steatosis-HCC. Because many of these genes could include those pertinent to the disease process of NASH versus steatosis, we further analyzed the differential gene expression in HCCs as compared to adjacent non-tumorous livers, and then compared cancer-related expressional changes between the two groups of HCCs.

- Relating to the human RNA seq data presented in figure 4, it is not usual to see RNA seq data filtered for a small set of genes but instead to analyze the entire transcriptome. I am concerned that the statistics employed (simple paired t-tests) do not account for multiple hypothesis testing. If the data is being presented to simply verify data from mice, then it would seem like qPCRs would have been more powerful.

RESPONSE:

We verified the genes identified in mouse models (figure 3c) in the 17 human NASH-HCC samples by RNA seq. The validated human RNA seq data presented in the

former figure 4 has been revised to a heat map (now figure 4a). Multiple hypothesis testing has been added to correct paired t-tests. Furthermore, we have added qPCR validation on the identified genes of *CFD*, *ALDH18A1*, *CHKA*, *DDIT3*, *ITGA6*, *PSPH* and *SQLE* in an additional cohort of 12 paired NASH-HCC human tissue samples (figure 4b). The new data and information have been added to the corresponding Method and Result sections as well as figure legend as following:

Page 8, line 24: False discovery rate (FDR) was adjusted using the Benjamini-Hochberg method for RNA seq data.

Page 11, line 27: Differential expression of seven of these genes (*CFD*, *ALDH18A1*, *CHKA*, *DDIT3*, *ITGA6*, *PSPH* and *SQLE*) was replicated in another set of 12 paired NASH-HCCs and adjacent non-tumor livers by RT-qPCR (all $P < 0.05$ by paired t-test; Fig. 4b).

Page 30, line 8: Figure 4. Aberrant gene expression verified in human NASH-HCC samples as compared to adjacent non-tumor livers. (a) Shown are RNA levels in FPKM in 17 pairs of tumors compared to adjacent non-tumor livers from patients with NASH-HCC. P values by paired t tests and FDR adjusted by Benjamini Hochberg method are shown. Expression levels were normalized to the mean level of each gene among all samples shown. (b) Expression of seven genes was further validated to be significantly altered in 12 pairs of NASH-HCCs (T) compared to adjacent non-tumor livers (N) by RT-qPCR. Comparison by paired t tests.

- Please provide a description of the statistical analysis performed in figure 7c.

RESPONSE:

The description of statistical analysis has been added to Method section and legend of figure 7c:

Page 8, line 25: Mutation frequencies were compared by chi-square test.

Page 35, line 7: (c) Mutation frequency of *RYR1* gene was higher in HCCs with risk history of NAFLD than others without NAFLD (Chi-square test).

- Although the authors do highlight some of the novel findings of this study in the discussion, this reviewer feels like even more should be done to discuss the novel findings, similar findings, and findings that are not in agreement from previous related studies. For example, there is at least one study I found looking at gene expression in human steatosis and NASH that don't seem to be discussed and neither is the actual TCGA publication from 2017.

RESPONSE:

We appreciate the comments. We have added information on the novel findings, similar findings, and findings that are not in agreement from previous related studies

in Discussion (**Page 15, line 2 – Page 18, line 26**).

Minor points:

- *In methods for liver sections by 5 m do you mean 5 micron?*

RESPONSE:

Yes, it is 5 micron. '5 m' has been revised as '5 μ m' (page 5, line 30). Thank you.

- *In Figure 5a, "HFHC" is cropped*

RESPONSE:

It has been corrected. Thank you.

- *Page 11 lines 19 and 20 why were the genes particular interest of "particular interest"? Also, the use of "etc" in gene lists could be somewhat confusing if a reader happened to think that was a gene symbol.*

RESPONSE:

These have been corrected accordingly. Thank you.

'particular interest' has been deleted and 'etc' has been revised as 'and others' to avoid confusion as following:

Page 11, line 9 (revised version): Genes commonly upregulated in both NASH- and SS-HCCs when compared to adjacent non-tumorous liver (such as *Cdh1*, *Pak1*, *Sqle*, and others) may be important in hepatocarcinogenesis. We also identified genes aberrantly expressed only in NASH-HCCs, such as down-regulation of *Cfd* (complement factor D, encoding adipsin), and up-regulation of *Ddit3*, *Itga6* and others (**Fig. 3c**).

- *Please provide scale bars on images in 1f1.*

RESPONSE:

Scale bars have been provided on images in 1f1.

- *For lung mets can you include a supplementary figure?*

RESPONSE:

Representative images of lung metastasis from a HFHC-fed mouse have been included in Supplementary Figure 1.

Reviewers' comments:

Reviewer #1 (Remarks to the Author):

The authors answered my concerns in a satisfactory way. The paper is now seriously improved.

Reviewer #2 (Remarks to the Author):

Authors addressed my concerns.

Reviewer #3 (Remarks to the Author):

Liang JQ, et al. analyzed transcriptomic data and whole exome sequencing in a mice model of DEN carcinogenesis with high fat diet together with high cholesterol diet. They validated their discovery in a small cohort of human HCC. I have some comments, especially in the way to analyze whole exome sequencing and identify new driver genes:

1. Did the authors observed an expression (mRNA level) of the main genes that they considered as driver after analysis of whole exome sequencing (such as RYR1, RYR2, CACNA1H) in mice and human non-tumor liver and in mice and human HCC ? A gene not expressed in the liver could not be a driver in liver carcinogenesis
2. The authors identified more somatic mutations in HCC from HFHC mice compared to HF mice. However, more mutations may be related to the more advanced stages of HCC analyzed in HFHC mice and not only related to specific carcinogenic events
3. In my mind, the numbers of tumor analyzed in mice and also in human is insufficient to identify correctly new driver genes. For example, mutations in gene observed by the authors could be due to stochastic mutation in a passenger gene (due to the large size of the genes for example) and not a true mutations in driver genes. Some algorithm have been used to identify new driver genes such as MutSigCV. The authors should use this algorithm.
4. I'm confused by the definition of "multiple small" and "multiple large" to define HCC in mice (Figure 1G). It is not objective.
5. It will be interesting to validate transcriptomic results observed in "house" human NASH HCC in NASH HCC from TCGA
6. The authors only used mutation of RYR1 of the TCGA data set to perform statistical analysis that compare NASH HCC versus non NASH HCC with a p value at 0.04 (figure 7C). However, if the authors put all NASH HCC together (home series + TCGA series) to compare with non-NASH HCC of TCGA data the differences will not be probably significant in my mind. This association (RYR1 with NASH HCC) seems not totally robust

Response to the comments of Reviewer 3 in relation to the manuscript:

MS# NCOMMS-17-31327-A, “Dietary cholesterol promotes steatohepatitis-related hepatocellular carcinoma by inducing aberrant gene expression in metabolism and mutations in calcium signaling”

COMMENTS are written in *italics* and RESPONSES in normal text.

Reviewer #3 (Remarks to the Author):

Liang JQ, et al. analyzed transcriptomic data and whole exome sequencing in a mice model of DEN carcinogenesis with high fat diet together with high cholesterol diet. They validated their discovery in a small cohort of human HCC. I have some comments, especially in the way to analyze whole exome sequencing and identify new driver genes:

1. Did the authors observed an expression (mRNA level) of the main genes that they considered as driver after analysis of whole exome sequencing (such as RYR1, RYR2, CACNA1H) in mice and human non-tumor liver and in mice and human HCC ? A gene not expressed in the liver could not be a driver in liver carcinogenesis

RESPONSE:

We appreciate the comments. All the main potential driver genes after analysis of whole exome sequencing (such as RYR1, RYR2, CACNA1H) are expressed in mice and human non-tumor liver and in mice and human HCC. Expression levels of RYR1, RYR2 and CACNA1H, as well as known cancer drivers MTOR and SDK1 as reference, are summarized below:

Arbitrary unit/microarray	Ryr1		Ryr2		Cacna1h		Mtor		Sdk1	
HFHC-mice (n=5)	Tumor	Normal	Tumor	Normal	Tumor	Normal	Tumor	Normal	Tumor	Normal
mean	7.78	7.73	6.50	8.24	16.50	15.39	139.62	141.71	57.63	45.59
SD	3.78	3.12	0.85	1.57	16.79	11.68	24.92	33.83	72.88	53.46
FPKM/HiSeq	RYR1		RYR2		CACNA1H		MTOR		SDK1	
In-house NASH-HCC (n=17)	Tumor	Normal	Tumor	Normal	Tumor	Normal	Tumor	Normal	Tumor	Normal
mean	0.0134	0.0108	0.0424	0.0140	0.6289	0.3870	4.8995	1.9497	0.0304	0.0372
SD	0.0151	0.0132	0.1516	0.0165	1.3812	0.4106	5.5783	1.6875	0.0677	0.0395
Log2(norm_count+1)/HiSeq	RYR1		RYR2		CACNA1H		MTOR		SDK1	
TCGA HCC (n=50; paired samples only)	Tumor	Normal	Tumor	Normal	Tumor	Normal	Tumor	Normal	Tumor	Normal
mean	3.550	3.469	2.973	2.613	7.835	8.590	10.628	10.43	4.414	4.351
SD	1.993	1.118	1.619	1.102	2.932	1.453	0.864	0.532	2.232	0.789

2. The authors identified more somatic mutations in HCC from HFHC mice compared to HF

mice. However, more mutations may be related to the more advanced stages of HCC analyzed in HFHC mice and not only related to specific carcinogenic events

RESPONSE:

We agree with the concern that more mutations may be related to the more advanced stages of HCC analyzed in HFHC mice. We have added this information in the revised Discussion as following:

Page 17, line 5: Oxidative DNA damage has been implicated in hepatocarcinogenesis in human NASH,⁵⁴ and inflammation has also been associated with DNA lesioning. Therefore, mutational difference in HCCs between HFHC- and HF-fed mice might be attributable to cholesterol-associated NASH development involving more severe inflammation. The increased number of mutations in HFHC tumors may also be partially due to more advanced stages of HCC or multiclonality as spatial genetic divergence has been observed in HCC.⁵⁵

3. In my mind, the numbers of tumor analyzed in mice and also in human is insufficient to identify correctly new driver genes. For example, mutations in gene observed by the authors could be due to stochastic mutation in a passenger gene (due to the large size of the genes for example) and not a true mutations in driver genes. Some algorithm have been used to identify new driver genes such as MutSigCV. The authors should use this algorithm.

RESPONSE:

We appreciate the comments and suggestion. As MutSigCV seems to be applied for large cohort of hundreds of samples, no significantly mutated genes is found in our 17 in house samples using MutSigCV probably due to the limited sample size. In addition, using MutSigCV, only 26 genes are identified to be significantly mutated from 363 HCC cases in TCGA data set (Cancer Genome Atlas Research Network. *Cell* 2017). Our study was to use mouse models to identify molecular changes and then validate the findings in human samples. MutSigCV currently can not analyze mouse genome data. In this regard, we have put this point in the revised Discussion.

Page 17, line 12 (Discussion): Due to the small sample size of human NASH-HCC data available, it is currently not feasible to use stringent methods such as MutSigCV, which takes into account of background mutations and other confounding covariates to identify significantly mutated genes. Our study took the advantage of cross-species oncogenomics to identify genetic alterations in mouse models and then validated such changes in human samples so as to identify the potential key genes associated with human NASH-HCC. We identified 21 recurrently mutated genes in mouse NASH-HCC models and further validated in human NASH-HCC. Key replicated genes included the mutational cancer drivers *MTOR* and *SDK1*, and three calcium signaling-related genes *RYR1*, *CACNA1H* and *RYR2*. Of note, *MTOR* has been determined to be an HCC driver gene using MutSigCV,⁵⁶ and was identified to be one of the most important genes associated with NASH-HCC in this study, suggesting the special significance of *MTOR* mutations in this subtype of HCC. Whether other genes

identified in this study are true driver genes in NASH-HCC need further validation in a larger cohort of samples as well as functional investigation.

4. *I'm confused by the definition of "multiple small" and "multiple large" to define HCC in mice (Figure 1G). It is not objective.*

RESPONSE:

Sorry for such confusing terms. We have now added the definition of "multiple small" and "multiple large" in the legend of Figure 1G as following:

Page 27, line 12: (g) HCC incidence, number and size in NC-, HF- and HFHC-fed mice after DEN injection. Tumor number: few<5; multiple≥5. Tumor size: small, all<5 mm³; large, at least one ≥5 mm³.

5. *It will be interesting to validate transcriptomic results observed in "house" human NASH HCC in NASH HCC from TCGA*

RESPONSE:

We appreciate this advice. We have evaluated the TCGA dataset. There are currently 371 human HCCs with RNA-seq data from TCGA dataset and 50 of them have paired adjacent non-tumor tissues. Among the 50 paired HCC cases, only 3 cases are NASH HCC. Thus, a validation is not able to be performed in NASH HCC from TCGA.

6. *The authors only used mutation of RYR1 of the TCGA data set to perform statistical analysis that compare NASH HCC versus non NASH HCC with a p value at 0.04 (figure 7C). However, if the authors put all NASH HCC together (home series + TCGA series) to compare with non-NASH HCC of TCGA data the differences will not be probably significant in my mind. This association (RYR1 with NASH HCC) seems not totally robust*

RESPONSE:

Considering potential batch effect, we also compared NASH HCC and non-NASH HCC of TCGA data set only. To avoid confusion, we have now added the non-significant p value to **Figure 7c**. We have also added the following description to Results section and Figure legend (for this comment as well as comment #3):

Page 14, line 20 (Results): Eight genes showed significantly higher mutation frequencies in NASH-HCCs (the combined cohort (n=37) or TCGA cohort (n=20)) as compared to other HCCs not complicating NASH/NAFLD (TCGA, n=353), and two of them encode calcium channel proteins (RYR1 and CACNA1H) (Fig. 7c).

Page 36, line 7 (legend of Figure 7): (c) Mutation frequencies of eight genes were higher in HCCs with risk history of NAFLD than others without NAFLD (Chi-square test). *P<0.05 and **P<0.001 for all NAFLD-HCCs (in-house and TCGA, n=37) versus other HCCs; #P<0.05 for NAFLD-HCCs from TCGA (n=20) versus other HCCs.

REVIEWERS' COMMENTS:

Reviewer #3 (Remarks to the Author):

The authors adequately answer to my comments